# LESS IS NOT WORSE: EFFECTIVE REASONING WITHOUT COMPLETE REASONING TRACES

## ABSTRACT

Large language models (LLMs) often produce lengthy reasoning traces with substantial token redundancy. While reasoning processes are widely adopted to tune LLMs as a post-training regime, it has been underexplored whether LLMs truly learn from the complete trajectory, particularly in supervised fine-tuning (SFT). We argue that, for mid-size LLMs commonly trained with SFT for reasoning, using full reasoning trajectories may harm performance because their limited capacity increases susceptibility to redundant intermediate steps. To investigate, we first analyze the redundancy in thinking trajectories through attention maps and controlled token-removal studies, both of which show that intermediate tokens contribute minimally to reasoning quality. Our analyses suggest that the most redundant segments typically appear in the middle of reasoning traces, whereas the earlier and later segments are crucial for generating high-quality final outcomes. We further posit that avoiding redundant intermediate information leads to exploiting the capability of LLMs to infer concise and coherent intermediate steps by utilizing the known start and end points. Based on the insights, we propose **MidCut**, a method that removes redundant middle steps during both training and inference. We demonstrate the effectiveness of **MidCut** in two scenarios for LLM reasoning: (1) SFT trained on s1K and OpenThoughts datasets for reasoning; and (2) decoding strategy for a test-time application.

## 1 INTRODUCTION

Learning to write for human learners may offer a helpful analogy for understanding how large language models (LLMs) (Grattafiori et al., 2024; Yang et al., 2024a) acquire reasoning skills. Learners benefit from carefully curated texts (*e.g.,* mentor texts), which are essential for building writing fundamentals (Kim et al., 2021; Culham, 2023; Shubitz, 2023). Yet, as learners advance, they increasingly dismiss misleading information as unhelpful, gaining little from it; they rather place more value on accurate and essential guidance, which drives more effective learning. A follow-up question would be: if provided only with essentials to learn yet disconnected parts, would they learn better (or even faster) and use prior knowledge to predict the missing pieces on their own?

LLMs have recently achieved strong results in reasoning tasks driven by large reasoning models (LRMs), such as the pioneer GPT-4o (Hurst et al., 2024) and various open-sourced initiatives (Yang et al., 2025; 2024b; Guo et al., 2025). The tasks span domains such as mathematical problem-solving and logical reasoning, which demand complex processes that are not known to be achievable solely through pre-training, mid-training, or alignment. LRMs are often trained through reward-based learning (Shao et al., 2024), or by leveraging machine-generated reasoning traces in a manner similar to how human learners are guided, to enhance reasoning capability. The reasoning traces, which represent the trajectory from a question to its answer, are informative for learning reasoning traces as supervisory signals; however, they often include redundant or unnecessary steps. Following the earlier analogy, one may ask whether a model trained only on essential sub-parts would fill the missing pieces in a trajectory with more plausible ones. However, even these are underexplored: identifying which parts of the reasoning trajectory are redundant and formulating a principle to exploit only the essential ones.

Existing studies examining the *thinking (reasoning) trajectory* of LRMs intentionally separated from their final answers to make the reasoning process explicit. Muennighoff et al. (2024) first introduced

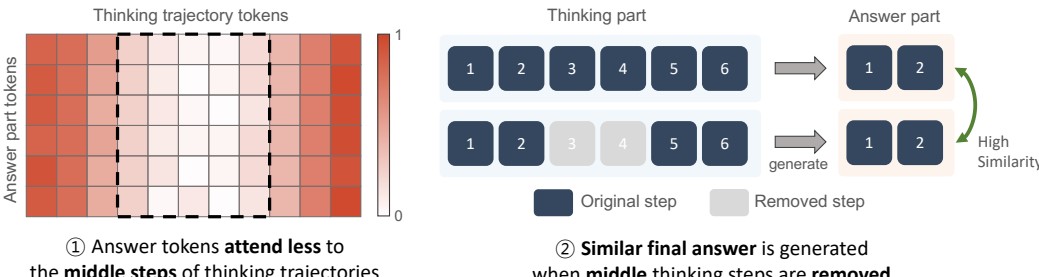

① Answer tokens **attend less** to the **middle steps** of thinking trajectories

② **Similar final answer** is generated when **middle** thinking steps are **removed**

Figure 1: **Redundancy may emerge in thinking trajectories.** Our insight is that a thinking trajectory often includes less informative traces, which we term redundancy. We preview our analyses for demonstration in §3: (left) low attention weights across thinking trajectories on an intermediate set of tokens; (right) removing them yields similar outputs.

the idea of using *entire* reasoning traces as guidance for supervised fine-tuning (SFT), and subsequent datasets such as (Guha et al., 2025) have further contributed reasoning traces for learning. Subsequently, however, several studies have noted that the thinking trajectories produced by LRMs are typically lengthy, less informative, and sometimes misleading (Ma et al., 2025b; Cuesta-Ramirez et al., 2025; Wu et al., 2025). Prior work further suggests that LLMs often already know the answer before generating a fully explicit reasoning trace (Lindsey et al., 2025). Literature (Ma et al., 2025b; Wang et al., 2025) demonstrates that problems can sometimes be efficiently solved without any explicit reasoning process, albeit with lower performance sometimes. These observations suggest that reasoning trajectories are important for solving complex problems, yet the fully explicit trajectory may not always be necessary. This raises a fundamental question: *do LLMs require the complete reasoning trajectory, and if not, which parts are non-essential?*

This paper focuses on SFT for mid-size LLMs using machine-generated reasoning traces (Muennighoff et al., 2024; Team, 2025; Labs, 2025; Guha et al., 2025). Our key insight is that such traces often contain substantial redundancy, particularly in intermediate steps; removing these redundancies enables more effective learning and yields significant performance gains due to the often-limited model capacity. To further investigate our insights, we first propose two systematic analyses: attention weight patterns and knockouts motivated by prior works (Ameisen et al., 2025; Clark et al., 2019; Madaan et al., 2023; Geva et al., 2023). We find that answer tokens place little attention on middle steps, and that removing these steps preserves answer quality, in contrast to removing the beginning or end, as shown in Figure 1. These findings indicate that **LLMs do not fully rely on the complete reasoning trajectory; intermediate steps are largely redundant and non-essential**. Motivated by these findings, we propose **MidCut**, a simple approach that trims intermediate steps of reasoning trajectories. **MidCut**-SFT leverages trimmed trajectories for more efficient supervised fine-tuning (SFT), while **MidCut**-Decoding skips redundant steps at inference to accelerate reasoning generation. Experiments demonstrate that both strategies improve efficiency while consistently maintaining strong performance across diverse reasoning tasks.

## 2 RELATED WORK

**Test-time scaling and consideration of overthinking.** Muennighoff et al. (2024) introduced a small curated dataset that is dubbed s1K having 1,000 reasoning trace examples satisfying difficulty, diversity, and quality, and *budget forcing* mechanism to control reasoning length at inference (*e.g.,* forcing early stop or appending "Wait" to prolong reasoning). Their model (*e.g.,* s1-32B) exhibits strong improvements in competition math questions (*e.g.,* AIME24) and shows a clear positive scaling-trend with test-time compute. After it, some recent works have challenged the assumption that longer reasoning chains always improve performance. Ghosal et al. (2025) analyzed test-time scaling and shows that beyond a certain point, longer chains can actually *hurt accuracy* due to "overthinking" rather than enhancing reasoning. Similarly, Hassid et al. (2025) demonstrated that, when multiple chains are sampled for the same question, the *shortest chain is often more reliable than the longest*, motivating inference-time strategies such as short-m@k and fine-tuning on shorter demonstrations. However, these approaches treat *length itself* as the key factor, without considering whether different segments of a chain (beginning, intermediate, and end) contribute unequally.

**Suppressing or replacing reasoning tokens for efficiency.** Another line of work explores omitting or substituting explicit 'thinking' tokens *to achieve efficient reasoning*. Ma et al. (2025a) found that reasoning models can remain competitive even when the *entire explicit reasoning block is skipped (NoThinking)*. Wang et al. (2025) showed that suppressing reflection-like markers such as "Wait" or "Alternatively" significantly shortens chains while maintaining performance. This demonstrates that not all generated tokens are necessary. Still, this strategy mainly targets *special filler tokens* rather than the broader redundancy of standard reasoning steps within the middle of a chain. Ringel et al. (2025) proposed a learned *continue-thinking token* that dynamically extends reasoning when needed. Both focus on whether to engage in any reasoning at all or on how to prolong it, but do not ask whether *some parts of the reasoning trajectory are inherently more redundant than others*.

**Reasoning compression and adaptive selectivity.** A similar line of work seeks to *compress* reasoning traces or *adaptively decide* how much to reason also for efficiency. At the trajectory level, Hou et al. (2025) pruned tokens *primarily considering sequence length* via reinforcement learning with an explicit token length limit, iteratively tightening the limit to shorten thoughts with minimal accuracy loss. Fan et al. (2025) proposed that *a light instruction model* drafts a high-level outline, and a reasoning model fills in details, which reduces generated tokens while maintaining accuracy, thus enabling difficulty-aware depth adjustment. Yuan et al. (2025) introduces a token-level compression framework that *scores reasoning tokens* and trains on compacted CoT while preserving accuracy. Lin et al. (2024) demonstrated at *pretraining* time that not all tokens are equally useful: they focus loss selectively on *high-utility tokens* (*i.e.,* useful and clean tokens in their terms), improving data efficiency – conceptually aligned with compression leading to faster training. These approaches either rely on token-importance scoring, budgeted RL, or guidance policies. By contrast, our approach offers a simple *region-level* recipe at training time: systematically remove the middle span of machine-generated chains during SFT, keeping the earlier and later spans of traces while discarding the redundant middle, without the need for auxiliary scorers or controllers.

**Our focus: to identify redundancy and to propose a simple solution to address it.** Our work is distinguished in three ways: (1) **region-level perspective**: we identify and omit a specific region rather than entire chains or individual tokens; (2) **training-time method**: unlike test-time heuristics, our method reshapes the training data itself; (3) **simple, low cost and reproducible**: our approach requires no auxiliary scoring modules and scales easily across large datasets. Finally, when recast as a question, while existing research asks *"How long should a trajectory be?"* or *"Which tokens should we selectively suppress?"*, this work asks *"Which parts of the trajectory are essential?"*

## 3    ANALYZING THINKING TRAJECTORIES

This section begins by presenting some preliminaries. We then investigate whether certain tokens within reasoning trajectories are redundant. We employ two complementary approaches: an attention-based analysis of how models process reasoning traces and a knockout-based analysis of how traces affect the quality of answer generation. Analysis overview is illustrated in Figure 1.

### 3.1    BACKGROUND

**Supervised fine-tuning (SFT)-based reasoning training.** Recent methods with carefully curated data: Sky-T1 (Team, 2025), s1K (Muennighoff et al., 2024), Bespoke-Stratos (Labs, 2025), OpenThoughts (Guha et al., 2025) suggest a promising paradigm, beyond RL-based reasoning training (Shao et al., 2024): reasoning models can be effectively trained with curated, machine-generated traces produced by an LRM, for example.

Formally, let $\mathcal{D} = \{(x_i, y_i, r_i)\}_{i=1}^{N}$ denote a altogether data of input query and answer pairs $(x_i, y_i)$ augmented with a reasoning trace $r_i$ generated by a LRM $T$ (*e.g.,* DeepSeek-R1 671B (Guo et al., 2025)). SFT then optimizes the parameters $\theta$ of a language model $M_\theta$ (usually smaller than $T$ *e.g.,* 3B-, 7B-, or 32B- scale) by minimizing the negative log-likelihood:

$$\mathcal{L}_{\text{SFT}}(\theta) = -\sum_{i=1}^{N} \log p_\theta(y_i, r_i \mid x_i), \tag{1}$$

where $r_i$ can be partially or fully included depending on the training recipe.

In this setting, high-quality reasoning traces provided by $T$ act as rich supervision signals, enabling smaller models $M_\theta$ to acquire strong reasoning ability without resorting to RL-based objectives (*e.g.,* GRPO (Shao et al., 2024)). This setup is particularly practical: large-scale traces are already well-formed by powerful teacher models, and SFT alone has been shown to yield competitive performance in smaller models.

**Problem setting.** We focus on SFT training with machine-generated reasoning traces (Muennighoff et al., 2024; Guha et al., 2025), applied to mid-scale models (*e.g.,* $\simeq$32B parameters). While RL often surpasses SFT (Chu et al., 2025), we presume that applying SFT is reasonable in this setting, both in terms of simplicity and expected performance. Muennighoff et al. (2024)'s findings implicitly supported the assumption: when large teacher models (*e.g.,* DeepSeek-R1 671B (Guo et al., 2025)) already produce high-quality reasoning traces, it is both realistic and effective to distill these traces into smaller models through SFT. Furthermore, since *SFT typically precedes RL methods* in RLHF procedures (Ouyang et al., 2022; Liu et al., 2024a; Achiam et al., 2023; Groeneveld et al., 2024) or is employed successively (OLMo et al., 2024; Yang et al., 2025), understanding the mechanics of SFT in relation to reasoning can give valuable insights. Therefore, we believe it would be valuable to study *how reasoning traces can be more effectively exploited for SFT training*.

**Our insight** is threefold: (1) redundancy (within reasoning chains of thinking trajectories) makes full traces difficult to learn with FT, particularly for models with limited capacity; (2) intermediate steps in reasoning traces are often redundant; and (3) removing them allows the model to focus on filling them with more relevant content to reach the final answer. Grounded in our insights, we conjecture that eq. (1) would become increased when some parts of reasoning trajectories are less informative or redundant. Before studying how redundancy affects the empirical behavior of LLMs, we first analyze redundancy in thinking trajectories.

## 3.2 ANALYSES

Figure 2 illustrates both example instances and the overall structure of reasoning trajectories. Such trajectories typically proceed from problem definition, through exploratory reasoning, to final answer consolidation. To investigate the functional roles of different parts in reasoning trajectories, we examine representative examples across reasoning tasks. Intriguingly, as similarly noted in prior work (Guo et al., 2025; Ma et al., 2025b; Cuesta-Ramirez et al., 2025; Wu et al., 2025), we often observe redundant traces (*e.g.,* repeated checks, backtracking, or unnecessary elaborations) in the middle of a trajectory. We now systematically investigate whether such human-perceived redundancy indeed matters for LLM.

**Attention weights analysis.** Following prior works (Ameisen et al., 2025; Clark et al., 2019; Madaan et al., 2023; Geva et al., 2023) that interpreted models using attention-based metrics, we analyze attention weights to investigate how different parts of thinking trajectories contribute to answer generation. We expect these patterns to reveal how models prioritize tokens during generation, offering insights into information flow within transformer architectures.

Figure 3 provides two complementary insights. First, the overall (*i.e.,* averaged across layers) trend in Figure 3(a) highlights **strong attention peaks at the beginning and ending of trajectories**, whereas intermediate steps are weakly attended. This suggests that intermediate steps generally contribute less to answer generation, indicating that these tokens could be redundant within the overall reasoning trajectory. Second, the layer-specific patterns in Figure 3(b–e) reveal a progression across the model depth: very early layers (pattern 1) attend broadly without clear localization, early layers (pattern 2) shift focus toward candidate answer tokens, intermediate layers (pattern 3) balance attention between early problem-definition and later reasoning steps, and final layers (pattern 4)

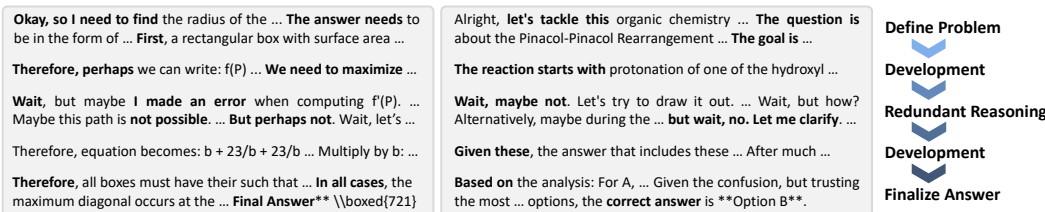

Figure 2: Examples of reasoning trajectories illustrating the overall reasoning process.

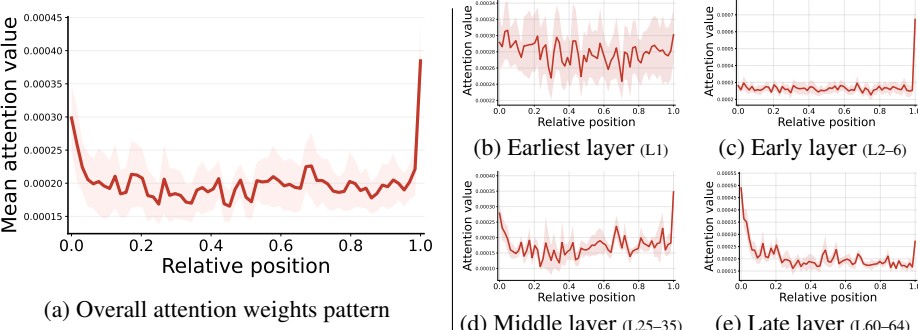

(a) Overall attention weights pattern

(b) Earliest layer (L1)

(c) Early layer (L2–6)

(d) Middle layer (L25–35)

(e) Late layer (L60–64)

Figure 3: **Averaged 1D attention weights across reasoning trajectories** during answer generation. (a) The overall average highlights strong attention peaks at the beginning and end of trajectories. (b–e) Layer-specific patterns reveal a progression: very early layers attend broadly, early layers shift their focus toward candidate answers, intermediate layers jointly reference problem-definition and reasoning steps, and final layers concentrate on the beginning to format and consolidate the answer.

concentrate strongly on the beginning to format and consolidate the final answer. Taken together, these findings suggest how information flow evolves through the network and provide evidence that intermediate reasoning steps are deemphasized as generation progresses toward the later layer.

**Attention knockout analysis.** We further apply the attention knockout technique (Geva et al., 2023) for our analysis, which masks attention links to probe the importance of specific trajectory segments. Specifically, we truncate the beginning (*e.g.,* the first 0–10% of tokens), the intermediate (*e.g.,* a centered span such as 45–55%), or the ending (*e.g.,* the last 90–100%) and compare the resulting answers against those from the full trajectories. This allows us to retest which part of reasoning trajectories are causally important for answer generation. Additional details on the experimental setup and evaluation metric are provided in Appendix B.1.

Based upon the observational insights from attention analysis, Figure 4 shows the average results over samples across different knockout ratios. In the results, **removing the intermediates consistently yields the highest similarity** to answers generated with the full trajectory. In contrast, removing the beginning and ending parts results in lower similarity, suggesting that these segments contain critical information for formulating correct answers. This provides quantitative evidence for our hypothesis that intermediate steps are often redundant and do not substantially contribute to final answer quality.

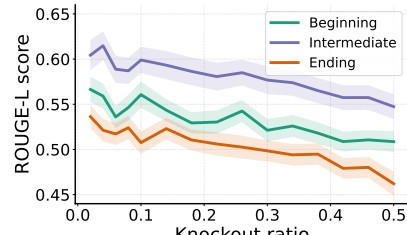

Figure 4: **ROUGE-L under attention knockouts** for each segment of reasoning traces. See more in Figure B.1.

> **Takeaway from** §3. **1** *Beginning and ending segments matter most, intermediate less:* Both attention and knockout analyses show that LLMs assign less importance to intermediate parts. **2** *Answer-first, then question-refinement:* attention patterns reveal an information flow in which models first lean on the final answer and then revisit the question for refinement.

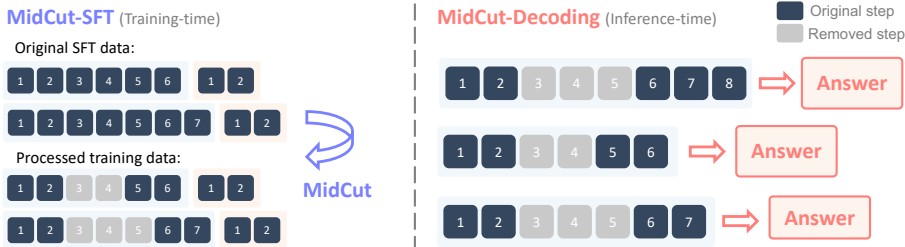

Figure 5: Overview of **MidCut**-SFT and **MidCut**-Decoding methodology. **MidCut**-SFT processes training data by removing middle parts from thinking trajectories, while **MidCut**-Decoding applies the same principle during inference to reduce computational overhead.

# 4 TRIMMING REASONING TRACES TO ENHANCE TRAINING AND DECODING

Based on the knowledge in §3, which reveals redundancy in the middle parts of thinking trajectories, we introduce **MidCut**, a method for removing intermediate steps from reasoning trajectories. As illustrated in Figure 5, we present two applications of our **MidCut** methodology: **MidCut-SFT** is a preprocessing method applied to LRM-generated reasoning trajectories in SFT datasets. This approach removes intermediate parts of the trajectory while preserving the beginning and ending segments, which are believed to contain the essential setup of the problem and the conclusion. **MidCut-Decoding** applies the same principle during inference, truncating intermediate reasoning steps after the thinking phase to reduce computational overhead and generation latency without compromising the quality of final answers. In both applications, our method targets the redundant middle parts identified in our analysis while preserving critical components at the trajectory boundaries.

## 4.1 MIDCUT-SFT: MIDCUT FOR SUPERVISED FINE-TUNING

**The proposed method.** **MidCut**-SFT follows a simple yet effective approach: given reasoning trajectories for training, we remove the intermediate traces. Specifically, we preserve the early and late segments of the reasoning trajectories based on predefined thresholds, either by step count or token count. Steps are defined by splitting trajectories at double newline characters ($\backslash$n$\backslash$n), represented by natural reasoning breakpoints.

We explore several variants of this filtering approach: (1) Step-level filtering: Remove intermediate steps while preserving the first and last $n$ steps of the trajectory. (2) Token-level filtering: Remove intermediate tokens while preserving the first and last $k$ tokens, regardless of step boundaries. (3) Length-proportional filtering: We remove a percentage of steps proportional to the original trajectory length, specifically removing $m\%$ of the total steps from the middle part. This approach scales the removal amount based on trajectory complexity. (4) Similarity-based filtering: We compute Jaccard similarity between each step and the preceding 5 steps of the trajectory. Steps exceeding a predefined similarity threshold are filtered out to reduce redundancy, targeting repetitive reasoning patterns. Further details on the truncation segments and parameter settings are provided in Appendix C.3.

**Comparison methods.** To evaluate the effectiveness of our method, we compare it to the setting that uses the original SFT dataset without any processing (referred to as "Base" throughout the paper), as well as to several alternative trajectory-reduction methods. These methods include: (1) Random Step Selection: we randomly sample $2n$ steps from the original trajectory, preserving the

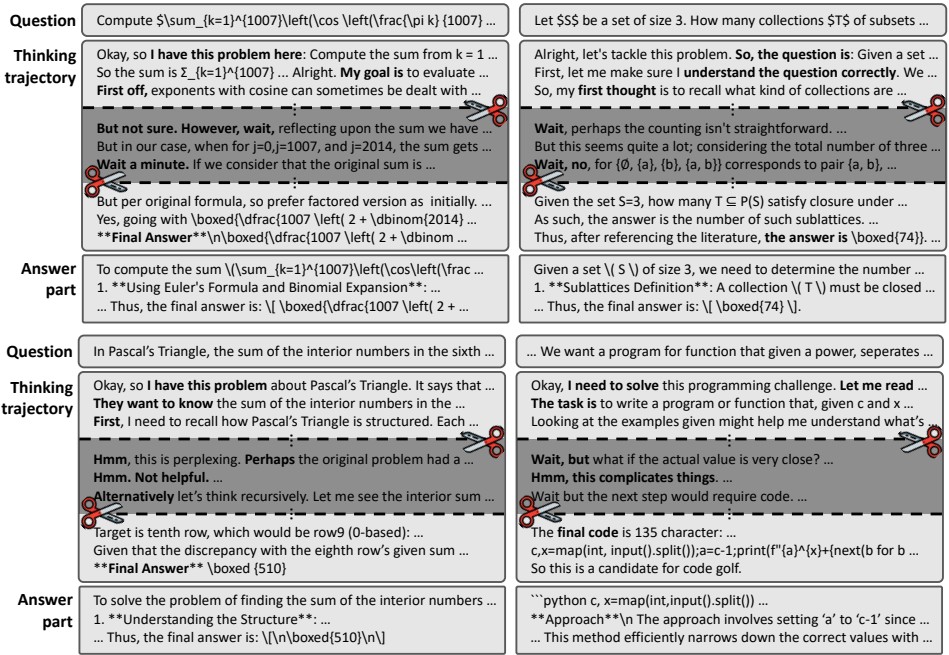

Figure 6: Examples of **MidCut**-SFT applied to (top) s1K-1.1 and (bottom) OpenThoughts3datasets.

Table 1: **Main comparison results** of `MidCut`-SFT on AIME24, GPQA-D, and MATH across different models and datasets. Positive relative changes are shown in blue, negative in red, and the best results for each task are highlighted in bold. "Base" refers to the model trained on the original SFT dataset without any modification. "Ours" denotes the step-level MidCut-SFT, and additional results for other variants are provided in Table 4 and Table C.2

| Method | AIME24 | | GPQA-D | | MATH | | Average | |
|---|---|---|---|---|---|---|---|---|
| | Value | Δ | Value | Δ | Value | Δ | Value | Δ |
| **Qwen2.5-32B-Instruct & s1K-1.1** | | | | | | | | |
| Base | 0.6444 | – | 0.6195 | – | 0.9413 | – | 0.7351 | – |
| LLM-based | 0.3333 | -48.28% | 0.5791 | -6.52% | 0.8900 | -5.45% | 0.6008 | -18.25% |
| Random | 0.5667 | -12.07% | 0.6162 | -0.54% | **0.9447** | +0.35% | 0.7092 | -3.53% |
| Ours | **0.6889** | +6.90% | **0.6229** | +0.54% | 0.9440 | +0.28% | **0.7519** | +2.29% |
| **Qwen3-8B-Base & s1K-1.1** | | | | | | | | |
| Base | 0.4222 | – | 0.5741 | – | **0.9210** | – | 0.6391 | – |
| LLM-based | 0.1556 | -75.85% | 0.4747 | -23.37% | 0.8233 | -12.52% | 0.4845 | -34.10% |
| Random | 0.2833 | -32.89% | **0.5741** | +0.00% | 0.8885 | -3.53% | 0.5820 | -8.94% |
| Ours | **0.4444** | +5.26% | **0.5741** | +0.00% | 0.9160 | -0.54% | **0.6448** | +0.90% |
| **Qwen3-8B-Base & OpenThoughts3-100K** | | | | | | | | |
| Base | 0.5000 | – | 0.5387 | – | 0.9347 | – | 0.6578 | – |
| Random | 0.4667 | -6.67% | 0.5312 | -1.40% | 0.9254 | -1.00% | 0.6411 | -2.54% |
| Ours | **0.5222** | +4.44% | **0.5640** | +4.70% | **0.9387** | +0.43% | **0.6750** | +2.61% |
| **Qwen3-4B-Base & OpenThoughts3-100K** | | | | | | | | |
| Base | 0.3667 | – | 0.4865 | – | **0.9160** | – | 0.5897 | – |
| Random | 0.3444 | -6.08% | 0.4680 | -3.80% | 0.9127 | -0.36% | 0.5750 | -2.49% |
| Ours | **0.4000** | +9.08% | **0.4899** | +0.70% | 0.9113 | -0.51% | **0.6004** | +1.81% |

same amount of content but without structural consideration. (2) Single-end Preservation: instead of preserving both beginning and end parts, we retain either the first $2n$ steps or the last $2n$ steps, maintaining the same total preserved length as `MidCut`-SFT. (3) LLM-based Compression: we use external LLMs – Claude Sonnet[1] (Anthropic, 2025) and Gemini (`gemini-2.5-flash`) (Comanici et al., 2025) – to compress thinking trajectories while preserving essential reasoning content. In the main text we report results with Claude, while detailed results with both Claude and Gemini are provided in Appendix C.4. (4) Perplexity-based Step Filtering: we utilize perplexity to filter out steps with high perplexity or with extreme values (either high or low) (Xie et al., 2023; Li et al., 2024b;a). (5) LS-Mixture SFT: as proposed in (Yu et al., 2025), where the authors incorporate LLM-based compression into a mixture-style SFT pipeline. Note that methods (3)–(5) require additional computational resources to reduce trajectories, while our method and (1)–(2) do not.

**Experimental settings.** We conduct experiments on two reasoning SFT datasets: s1K-1.1 (Muennighoff et al., 2024) and OpenThoughts3-1.2M (Guha et al., 2025). For OpenThoughts3, we subsample 100K examples from the full 1.2M corpus, and further restrict to instances where both the begin-of-think (`<think>`) and end-of-think (`</think>`) tokens are present. On s1K-1.1, we finetune `Qwen2.5-32B-Instruct` and `Qwen3-8B-Base` for 5 epochs, while on OpenThoughts3 we fine-tune `Qwen3-8B-Base` and `Qwen3-4B-Base` for 1 epoch. More detailed settings are described in Appendix C.3. We also utilize models from the LLaMa family (Grattafiori et al., 2024), and the corresponding details are reported in Appendix C.4. We mainly use AIME24, GPQA-D, and MATH to evaluate reasoning performance; TruthfulQA (Lin et al., 2022), MMLU (Hendrycks et al., 2021), HellaSwag (Zellers et al., 2019), and WinoGrande (Sakaguchi et al., 2021) to assess general-purpose language performance; and LiveCodeBench (Jain et al., 2024) and CodeElo (Quan et al., 2025) to evaluate code generation performance.

**Experimental results.** Table 1 shows the overall results that `MidCut`-SFT improves accuracy while using substantially fewer training tokens, with trimming the middle of thinking trajectories even outperforming training on full trajectories. Specifically, training on s1K-1.1 uses 21.9% fewer tokens and training on OpenThoughts3 uses 18.1% fewer tokens compared to full-

---

[1] We utilize the `claude-sonnet-4-20250514` version of Claude Sonnet 4.

Table 2: **Comparison of resource-free filtering strategies** in Table 1 on s1K-1.1. Each value represents the average performance on AIME24, GPQA-Diamond, and MATH. "Base" refers to the model trained on the original SFT dataset without any modification. Relative changes from the base are shown in parentheses below each value.

| Model | Base | Ours | Single (begin) | Single (end) | Random |
|-------|------|------|----------------|--------------|--------|
| Qwen2.5-32B | 0.7351 | **0.7519** | 0.7367 | 0.7183 | 0.7092 |
|  | – | (+2.29%) | (+0.22%) | (-2.30%) | (-3.53%) |
| Qwen3-8B | 0.6391 | **0.6448** | 0.6099 | 0.5698 | 0.5820 |
|  | – | (+0.90%) | (-4.56%) | (-10.82%) | (-8.94%) |

Table 3: **Comparison with resource-intensive filtering strategies** on s1K-1.1 with Qwen2.5-32B-Instruct. Each value represents the average performance on AIME24, GPQA-Diamond, and MATH. "Base" refers to the model trained on the original SFT dataset without any modification. Relative changes from the base are shown in parentheses below each value.

| Base | Ours | LLM-based | Perplexity-extreme | Perplexity-high | LS-Mixture SFT |
|------|------|-----------|--------------------|-----------------|----------------|
| 0.7351 | **0.7519** | 0.6008 | 0.7280 | 0.7443 | 0.7190 |
| – | (+2.29%) | (-18.27%) | (-0.97%) | (+1.25%) | (-2.19%) |

trajectory SFT. `MidCut`-SFT also yields practical benefits, reducing the wall-clock training time of `Qwen2.5-32B-Instruct` on s1K-1.1 by about 21%.

Table 2 and Table 3 show the extended comparison results on s1K-1.1 against alternative reductions, and further detailed results are provided in Appendix C.4. Approaches that preserve the original text but selectively trim segments generally achieve better performance than LLM-based compression. Among them, keeping both the beginning and ending while removing the middle consistently outperforms single-end truncation or random selection. This confirms that the location of truncation matters. On the other hand, LLM-based compression performs noticeably worse because it disrupts patterns of word usage and reasoning structure. Perplexity-based filtering and LS-Mixture SFT achieve comparatively strong performance, but neither surpasses ours. In addition, both require non-negligible computational resources. For example, the perplexity-based method alone takes more than 2 hours on a single H100 node, and LLM-based methods do not provide precise control over the compression ratio. Overall, these results support our hypothesis that intermediate steps are largely redundant and can even hinder effective learning if retained.

Table 4 reports ablations of `MidCut`-SFT. Step-level trimming, which preserves the beginning and ending segments at the semantic level, consistently outperforms token-level trimming. Length-proportional or similarity-based variants require additional complexity, but they do not surpass the simpler step-level `MidCut`. In particular, similarity-based filtering is less effective because it frequently misclassifies important early steps as redundant due to their high lexical overlap. This causes key definitions or setups to be dropped. These results indicate that step-level `MidCut`-SFT is both simple and effective, making it preferable to more complex alternatives.

Furthermore, Table 5 shows the effectiveness of `MidCut` in various datasets. In the case of general-purpose language performance, the base model without any reasoning-oriented SFT already demonstrates strong results. As commonly observed, domain-specific SFT tends to reduce performance on general-language benchmarks, leading to a noticeable degradation after training. However, the

Table 4: Comparison among **trimming variants** of `MidCut`-SFT on s1K-1.1. Each value represents the average performance on AIME24, GPQA-Diamond, and MATH. "Base" refers to the model trained on the original SFT dataset without any modification. Relative changes from the base are shown in parentheses below each value. "Len-prop." denotes length-proportional filtering.

| Model | Base | Step-level | Token-level | Len-prop. | Similarity |
|-------|------|-----------|-------------|-----------|------------|
| Qwen2.5-32B | 0.7351 | **0.7519** | 0.7264 | 0.7470 | 0.7392 |
|  | – | (+2.29%) | (-1.19%) | (+1.62%) | (+0.56%) |
| Qwen3-8B | 0.6391 | **0.6448** | 0.6162 | 0.6040 | 0.5994 |
|  | – | (+0.90%) | (-3.59%) | (-5.49%) | (-6.21%) |

Table 5: Comparison on various benchmarks using Qwen3-4B-Base and OpenThoughts3-100K. "Baseline" and "Base" denote to the original Qwen3-4B-Base model without any post-training (SFT) and the model trained on the original SFT dataset without any modification, respectively. "T-QA" and "LCB" refer to the "TruthfulQA" and "LiveCodeBench" benchmarks, respectively.

| | General language-centric | | | | | Code generation | |
|---|---|---|---|---|---|---|---|
| | T-QA-MC1 | T-QA-MC2 | MMLU | HellaSwag | WinoGrande | LCB | CodeElo |
| Baseline | 0.3709 | 0.5342 | 0.7319 | 0.5582 | 0.7143 | 0.0430 | 0.0332 |
| Base | 0.3256 | 0.4968 | 0.7253 | 0.5477 | 0.7040 | 0.3840 | 0.1197 |
| Ours | 0.3317 | 0.4965 | 0.7290 | 0.5486 | 0.7064 | 0.3847 | 0.1159 |

model trained with `MidCut`-SFT consistently maintains higher performance than the model trained with standard SFT. This indicates that removing redundant intermediate reasoning steps helps preserve general linguistic ability during domain-specific fine-tuning. In addition, the OpenThoughts3 dataset contains code-related instances. When evaluated on code-generation benchmarks, the model trained with `MidCut`-SFT achieves performance comparable to other approaches. These results collectively suggest that `MidCut`-SFT, which truncates redundant portions of the trajectories, does not compromise model quality across different evaluation domains.

## 4.2 MidCut-Decoding: MidCut for effective decoding

**Method.** `MidCut`-Decoding applies the middle-cutting principle during inference to reduce computational overhead. After the model completes its thinking trajectory, we cut the middle part of the trajectory before producing the final answer. We implement four ratios with {25, 33, 50, 75}% of the middle part while preserving beginning and ending segments. It reduces the computations required to generate the answer part and yields real speed gains in end-to-end evaluation. More details are reported in Appendix D.

Table 6: **Performance of MidCut-Decoding.**

| Method | Accuracy |
|---|---|
| Full trajectory | 0.6578 |
| `MidCut` 25% | 0.6574 |
| `MidCut` 33% | 0.6579 |
| `MidCut` 50% | 0.6581 |
| `MidCut` 75% | 0.6576 |

**Experimental results.** Table 6 reports `MidCut`-Decoding results on the SFT-trained model with OpenThought3. Compared to using the full reasoning trajectory, trimming 25–75% of the intermediate steps yields almost identical average performance. These findings indicate that `MidCut`-Decoding can substantially reduce inference costs for long reasoning traces while maintaining comparable task accuracy.

> **Takeaway from §4.** *Less is not worse, sometimes even better.*
> Removing redundant intermediate steps with `MidCut`-SFT improves reasoning performance, while inherently reducing training cost. Likewise, `MidCut`-Decoding sustains reasoning quality while improving efficiency in answer generation by removing redundant middle parts.

## 5 Discussion

This section initiates discussions by posing crucial questions that help us gain a deeper understanding of our work. We first validate our claim that training loss is indeed reduced, and subsequently discuss what our method achieves indeed and under which conditions it is effective.

**Does MidCut-SFT lead to a better convergence point?** Here, we evaluate whether our conjecture holds by examining the loss curves during training on OpenThoughts3, with and without `MidCut`-SFT. As shown in Figure 7, our method consistently achieves lower perplexity across training while using fewer tokens. This finding supports our claim in §3 that redundant intermediate steps not only prolong trajectories but also inflate the training loss, which presumably makes optimization harder for LLMs; thus, LLMs would converge to a suboptimal point. By trimming such redundancy, the model can allocate its capacity to more informative reasoning signals, which in turn facilitates more efficient and stable learning dynamics.

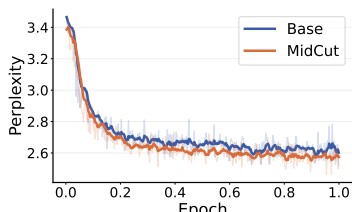

Figure 7: **Perplexity changes** when applying `MidCut`-SFT.

Figure 8: Token length changes.

Figure 9: Impact of **MidCut**-SFT on intermediate traces.

**What does MidCut-SFT do?** Building on the loss analysis, we next examine how **MidCut**-SFT alters the actual reasoning behavior of models. As shown in Figure 8, we observe a consistent pattern: the number of reasoning tokens with **MidCut**-SFT adjusts depending on the baseline trajectories without **MidCut**-SFT. When original trajectories are excessively long (>15k tokens), the output length decreases; for relatively shorter ones, token usage increases. This adaptive adjustment is further illustrated with the examples in Figure 9. In the upper row, where the baseline model produces unnecessarily lengthy and redundant output, **MidCut**-SFT suppresses such detours, enabling the model to reach a more accurate solution with fewer tokens. In the lower row, although both models arrive at the same intermediate equations, the baseline fails to solve the problem, whereas **MidCut**-SFT succeeds by more effectively completing the missing steps in the reasoning process. These observations suggest **MidCut**-SFT enhances performance by (1) reducing redundant reasoning patterns and (2) improving the model's ability to carry out intermediate reasoning when needed.

**When does MidCut-SFT work?** Figure 10 plots the relationship between general performance (MMLU-Pro and MMLU-redux) without any additional training and performance gain from applying **MidCut**-SFT. Each point corresponds to a different base model, with results shown for both s1K-1.1 and OpenThoughts3. The detailed experimental settings are provided in Appendix C.5. We find that the effectiveness of **MidCut**-SFT depends strongly on the original capability of the base model. For s1K-1.1, improvements appear only when the base performance exceeds roughly 60–65, whereas the larger

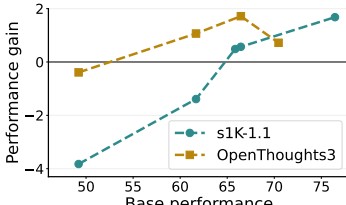

Figure 10: Performance gain by **MidCut**-SFT.

and more diverse OpenThoughts3 dataset shifts this threshold lower, allowing benefits to emerge even for weaker base models. This trend aligns with our earlier discussion: **MidCut**-SFT enhances the model's ability to complete intermediate reasoning steps, but this requires a sufficient baseline capacity. When models are too weak, they lack the ability to reliably infer the trimmed parts, and the method brings little to no benefit. By contrast, once a model has reached a moderate competence level, trimming redundant reasoning forces it to allocate capacity toward informative steps, thereby yielding measurable gains. This pattern closely parallels our analogy to human learners: while novices depend on full guidance, more advanced learners can benefit from streamlined instruction and effectively interpolate the missing reasoning by leveraging prior knowledge.

## 6 CONCLUSION

We have studied the necessity of complete reasoning trajectories and proposed **MidCut** for trajectory reduction. Through our analyses, we have shown that intermediate reasoning traces are often redundant. **MidCut**-SFT achieved the improved reasoning results further with efficiency, and **MidCut**-Decoding reduced inference costs without compromising quality. Beyond SFT and decoding, our method may also be integrated into reinforcement learning (RL) approaches such as GRPO, where reducing redundant reasoning trajectories could further enhance reasoning performance. We believe that **MidCut** could serve as an effective initialization for RL-based methods, and that the insights from our study may provide complementary guidance for their development. Overall, our simple yet effective method, which showcases how to handle lengthy reasoning trajectories without relying on any challenging restrictions, provides a promising research direction and paves the way for developing more efficient reasoning models in the near future.

**Reproducibility statement.** We performed all experiments using the s1K-1.1 and OpenThoughts3 datasets and publicly available LLMs from the Qwen series. Detailed hyperparameters and training

schedules are included in the Appendix, and all evaluations were repeated across multiple random seeds, with averaged results reported. These efforts ensure that our results can be readily reproduced.

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

# Appendix

This appendix provides additional experimental details and results that complement the main paper. Appendix A discusses the connection between our analysis and "Lost in the Middle" phenomenon. In Appendix B, we elaborate on the experimental settings and extended experimental results of our attention knockout analyses. Appendix C presents further implementation details and additional results of **MidCut**-SFT, including comparisons on the OpenThoughts3-100K dataset and results with alternative LLM-based compression methods. We also report extended results of Table 1 that complement the tables in the main text. Appendix D provides implementation details of **MidCut**-Decoding, specifying how the middle segments are truncated during inference. Finally, Appendix E clarifies the limited role of external LLMs in this work, restricted solely to writing polish and coding assistance, with no involvement in research ideation, implementation, or analysis.

## A  DISCUSSION ON ATTENTION WEIGHT ANALYSIS

Figure 3 shows a clear tendency in the attention pattern, suggesting a possible connection to the "Lost in the Middle" effect (Liu et al., 2024b). While there are some differences, there may be a shared connection. First, our analysis shows that LRMs pay less attention to the intermediate parts of reasoning trajectories, leading us to conclude that these segments contain many redundant steps that the model naturally deprioritizes. On the other hand, the "Lost in the Middle" phenomenon demonstrates that LLMs struggle to retrieve crucial information when it appears in the middle of a long context, indicating that even important content in this region receives insufficient attention.

Despite these differences, both observations point to a shared underlying pattern: LLMs generally assign insufficient attention to the middle region of long contexts. Specifically, in (Liu et al., 2024b), the authors explain that the "Lost in the Middle" effect is linked to the structure of SFT datasets for instruction tuning, in which task specifications and instructions are often placed at the beginning. Reasoning trajectories exhibit a similar structure: the problem setup and conclusions naturally appear at the beginning and end, while the middle often consists of decomposed but redundant reasoning steps. This distribution of information may cause LRMs to develop weaker attention to the middle, which in turn allows redundant middle content to proliferate during generation. Thus, while our finding differs in mechanism, redundancy-driven attention drop rather than attention-driven failure, the two phenomena may be linked through the underlying structure of the training data.

## B  DETAILS OF ATTENTION KNOCKOUT ANALYSIS

### B.1  EXPERIMENTAL SETTINGS

We delete specific trajectory segments and generate answers under these conditions to verify again whether thinking trajectories contain redundancy with respect to answer generation. By observing how answer quality changes under such interventions, we can assess whether particular segments play a causal role in generating answers. Our analysis is conducted on 100 problems from the GPQA-D dataset using the `s1.1-32B` (Muennighoff et al., 2024) model[2]. For each problem, a full thinking trajectory is first generated. Then, we remove three different segments: the beginning segment, an intermediate segment centered in the middle, and the ending segment. To systematically examine the effect of segment length, we vary the truncated ratio (e.g., 10%, 20%). For instance, with a 10% ratio the beginning corresponds to tokens 0–10%, the intermediate to 45–55%, and the ending to 90–100% of the trajectory. With a 20% ratio, the corresponding spans become 0–20%, 40–60%, and 80–100%, respectively. The model is subsequently prompted to generate answers from these truncated trajectories as well as from the full trajectory. Finally, textual similarity between answers from truncated and full trajectories is measured using Jaccard similarity (Manning, 2008) and ROUGE-L (Lin, 2004).

### B.2  ADDITIONAL RESULTS

Complementing the ROUGE-L (Lin, 2004) results in Figure 4 of the main paper, Figure B.1 reports Jaccard similarity (Manning, 2008) and BLEU score (Papineni et al., 2002) under the same knock-

---

[2] https://huggingface.co/simplescaling/s1.1-32B

out settings. The trends are closely similar to those of the ROUGE-L score. Across all knockout ratios (0–50%), removing the intermediate segment consistently yields the highest similarity between answer parts generated from truncated trajectories and their counterparts generated from the full trajectory. In contrast, removing the ending segment yields the lowest similarity and is even worse than removing the beginning. This aligns with the information flow revealed by the attention weight analysis (Figure 3). Models appear to anchor the answer first and then revisit the question for refinement, so truncating the final consolidation region most harms the generated answer parts.

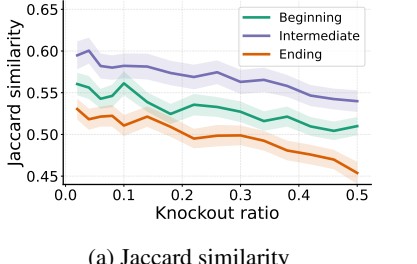 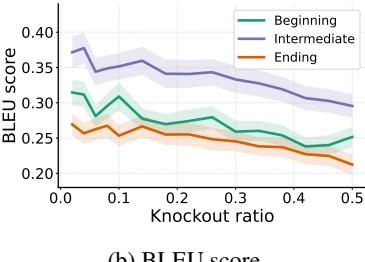

(a) Jaccard similarity            (b) BLEU score

Figure B.1: **(a) Jaccard similarity and (b) BLEU score** of generated answer parts after knockouts of thinking trajectories. They also show similar trends to Figure 4.

## C    DETAILS OF **MIDCUT**-SFT EXPERIMENTS

### C.1    SFT AND EVALUATION FRAMEWORKS

For the **s1K-1.1 experiments**, we conduct supervised fine-tuning using the s1 GitHub repository[3]. Evaluation relies on the built-in implementation of lm-evaluation-harness[4] within the same repository. The dataset is s1K-1.1[5].

For the **OpenThought3 experiments**, we follow the instructions of the OpenThought GitHub repository[6] and train models using LLaMA-Factory[7]. Evaluation is performed with lm-evaluation-harness[2]. The dataset is based on OpenThoughts3-1.2M[8], from which we randomly sample 100K examples containing both `<think>` and `</think>` tokens.

### C.2    LLM-BASED COMPRESSION METHOD

We also experiment with compressing the s1K-1.1 dataset using two external LLMs: Claude Sonnet (`claude-sonnet-4-20250514` version of Claude Sonnet 4) (Anthropic, 2025) and Gemini (`gemini-2.5-flash`) (Comanici et al., 2025).

To ensure that only the middle portion is shortened while respecting its local context, we operate on the thinking trajectory with a block-wise, context-aware paraphrasing pipeline. Each trajectory is split into "blocks" by "\n\n", and we iterate over non-overlapping windows of 10 consecutive blocks. For every window, we construct a prompt that exposes the three preceding and three following blocks as `[Context Before]` and `[Context After]`, while designating the 10-block window as `[Center Section]`. The LLM is instructed to compress the `[Center Section]` only, preserving the logical structure and final conclusion. We instruct the LLMs with a fixed paraphrasing prompt, which explicitly conditions on surrounding context while compressing only the center section. The full prompt is shown in the box below.

---

[3] https://github.com/simplescaling/s1
[4] https://github.com/EleutherAI/lm-evaluation-harness
[5] https://huggingface.co/datasets/simplescaling/s1K-1.1_tokenized
[6] https://github.com/open-thoughts/open-thoughts
[7] https://github.com/hiyouga/LLaMA-Factory
[8] https://huggingface.co/datasets/open-thoughts/OpenThoughts3-1.2M

---

**Paraphrasing prompt**

You will be given a step-by-step reasoning process written by a large language model.

Please paraphrase the **center section only**, while preserving the logical structure.
Try to reduce its length to approximately half of the original, but make sure to keep all essential reasoning steps and the final conclusion.
You may refer to the surrounding context for understanding, but do not modify them.

```
[Context Before]
```
{before}

```
[Center Section]
```
{center}

```
[Context After]
```
{after}

**Paraphrased (center section only):**

---

## C.3 IMPLEMENTATION DETAILS AND EXPERIMENTAL SETTINGS

In all variants of **MidCut**-SFT, the middle portion of the trajectory is removed after preserving both the beginning and ending segments. Concretely, for step-level filtering we preserve the first and last $n$ steps and remove the middle (total steps $- 2n$) steps; for token-level filtering we preserve the first and last $k$ tokens and remove the middle (total tokens $- 2k$) tokens; and for length-proportional filtering we remove $m\%$ of the total steps from the middle part. If the preserved length ($2n$ steps or $2k$ tokens) exceeds the total trajectory length, no truncation is applied.

For step-level filtering, we set $n{=}100$ for s1K-1.1 and $n{=}200$ for OpenThoughts3. We first chose $n{=}100$ to remove approximately 20% of the total tokens on s1K-1.1, and found that the preserved first and last steps naturally matched its average trajectory length (234 steps). Based on the same principle, we set $n{=}200$ for OpenThoughts3 (average 461 steps), which likewise corresponds to removing about 20% of the tokens. Step-level midcut is used as the default configuration, while further exploratory ablations (token-level with $k = 8000$ and length-proportional with $m = 0.15$) are carried out on s1K-1.1. For similarity-based **MidCut**, we experimented with different filtering criteria (max, min, and average Jaccard similarity against a threshold) and found that the best performance was obtained with an average threshold of 0.3 for Qwen3-8B-Base and a minimum threshold of 0.15 for Qwen2.5-32B-Instruct, which we report in the main results. For evaluation, we use AIME24, GPQA-D, and MATH benchmarks, generating with temperature 0.6. Each instance is evaluated three times, and we report the averaged results.

## C.4 ADDITIONAL RESULTS OF **MidCut**-SFT

Table C.1 compares the performance of **MidCut**-SFT on OpenThoughts3-100K against alternative reduction strategies. Consistent with the results on s1K-1.1 (Table 2), the step-level **MidCut**-SFT achieves the highest performance among all methods, yielding relative improvements of +2.61% on Qwen3-8B-Base and +1.81% on Qwen3-4B-Base compared to training on full trajectories.

Table C.2 presents detailed results by task, model, and method, complementing Table 1-4 and Table C.1. Overall, the results confirm that **MidCut**-SFT consistently achieves strong performance.

Table C.3 shows the results on the LLaMA family using OpenThoughts3-100K for SFT. We fine-tune the models for 1 epoch on the OpenThoughts3-100K dataset, following the same setting used for the Qwen-series models. **MidCut**-SFT maintains performance on LLaMA-3.2-3B-Instruct and provides a modest improvement on LLaMA-3.1-8B-Instruct (+1.34% on average). These results

suggest that **MidCut**-SFT generalizes beyond the Qwen-based models and remains effective across different model families without introducing performance degradation.

Table C.1: Comparison between **MidCut**-SFT (Step-level) and alternative truncation strategies on OpenThoughts3-100K. Each value represents the average performance on AIME24, GPQA-Diamond, and MATH. Relative changes from the base are shown in parentheses below each value.

| Model | Base | Random | Ours (Step-level) | Single (first) | Single (last) |
|-------|------|--------|-------------------|----------------|---------------|
| Qwen3-8B | 0.6578 – | 0.6411 (-2.54%) | **0.6750** (+2.61%) | 0.6216 (-5.50%) | 0.5986 (-9.00%) |
| Qwen3-4B | 0.5897 – | 0.5750 (-2.49%) | **0.6004** (+1.81%) | 0.5816 (-1.38%) | 0.5268 (-10.68%) |

## C.5 EXPERIMENTAL DETAILS OF FIGURE 10

To examine under which conditions **MidCut**-SFT is most effective, we analyze the correlation between the general task performance of the pretrained models and the performance gain of **MidCut**-SFT over full-trajectory SFT. For this experiment, we train `Qwen2.5-32B-Instruct`, `Qwen2.5-7B-Instruct`, `Qwen3-8B-Base`, and `Qwen3-4B-Base` on s1K-1.1, and `Qwen3-8B-Base` and `Qwen3-4B-Base` on OpenThoughts3-100K. The results are then compared to investigate how model capacity and initialization relate to the observed performance gains.

## D IMPLEMENTATION DETAILS OF **MIDCUT**-DECODING

The model first generates the entire thinking trajectory up to the point where it would normally begin producing the final answer. Before the answer part is generated, we truncate the central portion of the thinking trajectory according to a specified cutting ratio. Formally, given a trajectory of length $L$ tokens, we remove $\lfloor r \cdot L \rfloor$ tokens centered at position $L/2$, thereby preserving both the first $(1-r)/2$ and last $(1-r)/2$ fractions of the trajectory. For example, with $r=0.5$, the central 50% of tokens (i.e., from 25% to 75% of the trajectory) is removed. This ensures that the model generates its final answer after observing both the initial problem setup and the concluding consolidation region, while discarding potentially redundant intermediate reasoning.

The efficiency benefit of **MidCut**-Decoding naturally arises from its decoding behavior: for long-form reasoning tasks, our method shortens the effective context seen by answer tokens and downstream modules. For example, in GPQA Diamond settings, models typically generate around 8.1k tokens of intermediate reasoning. In a naive two-stage pipeline, this entire 8.1k token trace must be re-prefilled during the second pass under a naive decoding procedure. In contrast, **MidCut**-Decoding (33%) compresses this reasoning to roughly 5.4k tokens. This theoretically reduces the second-pass prefill computation by about 46% and also reduces the attention computation in answer generation (for around 0.5k tokens) to 68%, while maintaining the same answer quality. We also observed a 4.5% reduction in end-to-end evaluation time on GPQA Diamond using `Qwen2.5-32B-Base`, fine-tuned on s1K-1.1, in our environment, despite LLM-generation functions (e.g., the vLLM library (Kwon et al., 2023)) in Python being highly optimized, including attention computation accelerated by FlashAttention (Dao et al., 2022; Dao, 2024).

The main results in Table 6 are obtained with `Qwen3-8B-Base` trained on OpenThought3-100K, while additional results with `Qwen2.5-32B-Instruct` trained on s1K-1.1 are provided in Table D.1. These additional results reinforce the observation from the main text that trimming 25–75% of intermediate steps yields nearly identical average performance compared to using full trajectories, indicating that **MidCut**-Decoding can reduce inference costs without harming task accuracy.

## E THE USE OF LLMS

LLMs were primarily used for minor language editing, including adjustments to word choices and clarity. They played no role in the research design, analysis, interpretation, or manuscript preparation, and all scientific contributions are fully our own.

Table C.2: **Detailed performance of `MidCut`-SFT on three benchmarks (AIME24, GPQA-D, MATH).** Bold indicates the best performance in each benchmark, and underline indicates performance better than baseline. "Len-prop." denotes length-proportional filtering.

| Method | AIME24 | | GPQA-D | | MATH | | Average | |
|---|---|---|---|---|---|---|---|---|
| | Value | Δ | Value | Δ | Value | Δ | Value | Δ |
| **Qwen2.5-32B-Instruct & s1K-1.1** | | | | | | | | |
| Base | 0.6444 | – | 0.6195 | – | 0.9413 | – | 0.7351 | – |
| LLM-based (Claude) | 0.3333 | -48.27% | 0.5791 | -6.52% | 0.8900 | -5.45% | 0.6008 | -18.27% |
| LLM-based (Gemini) | 0.1778 | -72.41% | 0.4815 | -22.27% | 0.8273 | -12.12% | 0.4955 | -32.59% |
| Perplexity-extreme | 0.6000 | -6.90% | 0.6313 | +1.90% | **0.9527** | +1.21% | 0.7280 | -0.97% |
| Perplexity-high | 0.6444 | 0.00% | **0.6465** | +4.36% | 0.9420 | +0.07% | 0.7443 | +1.25% |
| LS-Mixture SFT (Mix) | 0.6000 | -6.90% | 0.6110 | -1.37% | 0.9460 | +0.50% | 0.7190 | -2.19% |
| Single (first) | 0.6222 | -3.44% | 0.6380 | +2.99% | 0.9500 | +0.92% | 0.7367 | +0.22% |
| Single (last) | 0.6000 | -6.90% | 0.6128 | -1.08% | 0.9420 | +0.07% | 0.7183 | -2.29% |
| Random | 0.5667 | -12.06% | 0.6162 | -0.53% | 0.9447 | +0.36% | 0.7092 | -3.52% |
| Ours | | | | | | | | |
|   Step-level | **0.6889** | +6.90% | 0.6229 | +0.55% | 0.9440 | +0.29% | **0.7519** | +2.29% |
|   Token-level | 0.6111 | -5.16% | 0.6195 | 0.00% | 0.9487 | +0.79% | 0.7264 | -1.17% |
|   Len-prop. | 0.6556 | +1.74% | 0.6380 | +2.99% | 0.9473 | +0.64% | 0.7470 | +1.62% |
|   Sim-based | 0.6444 | 0.00% | 0.6279 | +1.36% | 0.9453 | +0.42% | 0.7392 | +0.56% |
| **Qwen3-8B-Base & s1K-1.1** | | | | | | | | |
| Base | 0.4222 | – | 0.5741 | – | **0.9210** | – | 0.6391 | – |
| LLM-based (Claude) | 0.1556 | -75.85% | 0.4747 | -23.37% | 0.8233 | -12.52% | 0.4845 | -34.10% |
| LLM-based (Gemini) | 0.1111 | -73.68% | 0.4293 | -25.22% | 0.7813 | -15.16% | 0.4406 | -31.06% |
| First | 0.3667 | -13.13% | 0.5640 | -1.76% | 0.8990 | -2.39% | 0.6099 | -4.55% |
| Last | 0.2778 | -34.23% | 0.5387 | -6.17% | 0.8930 | -3.04% | 0.5698 | -10.80% |
| Random | 0.2833 | -32.94% | 0.5741 | +0.00% | 0.8885 | -3.53% | 0.5820 | -8.94% |
| Ours | | | | | | | | |
|   Step-level | **0.4444** | +5.26% | **0.5741** | +0.00% | 0.9160 | -0.54% | **0.6448** | +0.90% |
|   Token-level | 0.3667 | -13.13% | 0.5690 | -0.89% | 0.9130 | -0.87% | 0.6162 | -3.58% |
|   Len-prop. | 0.3333 | -21.00% | 0.5606 | -2.35% | 0.9180 | -0.33% | 0.6040 | -5.49% |
|   Sim-based | 0.3444 | -18.42% | 0.5539 | -3.52% | 0.9000 | -2.28% | 0.5994 | -6.22% |
| **Qwen3-8B-Base & OpenThought3-100K** | | | | | | | | |
| Base | 0.5000 | – | 0.5387 | – | 0.9347 | – | 0.6578 | – |
| First | 0.4000 | -20.00% | 0.5320 | -1.24% | 0.9327 | -0.21% | 0.6216 | -5.51% |
| Last | 0.4111 | -17.78% | 0.5135 | -4.68% | 0.8713 | -6.79% | 0.5986 | -9.00% |
| Random | 0.4667 | -6.67% | 0.5312 | -1.40% | 0.9254 | -1.00% | 0.6411 | -2.54% |
| Ours (Step-level) | **0.5222** | +4.44% | **0.5640** | +4.70% | **0.9387** | +0.43% | **0.6750** | +2.61% |
| **Qwen3-4B-Base & OpenThought3-100K** | | | | | | | | |
| Base | 0.3667 | – | 0.4865 | – | **0.9160** | – | 0.5897 | – |
| First | 0.3889 | +6.05% | 0.4646 | -4.50% | 0.8913 | -2.69% | 0.5816 | -1.38% |
| Last | 0.2889 | -21.20% | 0.4495 | -7.60% | 0.8420 | -8.08% | 0.5268 | -10.68% |
| Random | 0.3500 | -4.56% | 0.4748 | -2.43% | 0.8987 | -1.88% | 0.5745 | -2.49% |
| Ours (Step-level) | **0.4000** | +9.08% | **0.4899** | +0.70% | 0.9113 | -0.51% | **0.6004** | +1.81% |

Table C.3: **Performance of `MidCut`-SFT on the LLaMa-family.**

| Method | AIME24 | | GPQA-D | | MATH | | Average | |
|---|---|---|---|---|---|---|---|---|
| | Value | Δ | Value | Δ | Value | Δ | Value | Δ |
| **LLaMa-3.2-3B-Base & OpenThoughts3-100K** | | | | | | | | |
| Base | 0.0767 | – | 0.2965 | – | 0.6320 | – | 0.3350 | – |
| Ours (Step-level) | 0.0667 | -1.00% | 0.3384 | +4.19% | 0.6000 | -3.20% | 0.3350 | +0.00% |
| **LLaMa-3.1-8B-Base & OpenThoughts3-100K** | | | | | | | | |
| Base | 0.1767 | – | 0.3754 | – | 0.7440 | – | 0.4320 | – |
| Ours (Step-level) | 0.2000 | +2.33% | 0.3822 | +0.68% | 0.7540 | +1.00% | 0.4454 | +1.34% |

Table D.1: **Performance of `MidCut`-Decoding.** "Accuracy" denotes the averaged accuracy across AIME24, GPQA-D, and MATH.

| Model | Full | 25% | 33% | 50% | 75% |
|---|---|---|---|---|---|
| Qwen3-8B-Base | 0.6578 | 0.6574 | 0.6579 | 0.6581 | 0.6576 |
| Qwen2.5-32B-Instruct | 0.7420 | 0.7421 | 0.7420 | 0.7420 | 0.7416 |

