# OpenReview forum: "Less is Not Worse: Effective Reasoning Without Complete Reasoning Traces"
_ICLR.cc/2026/Conference — Submitted to ICLR 2026_

### Official Review · Reviewer_ZoqT · 2025-10-22

**Soundness:** 2
**Presentation:** 3
**Contribution:** 1
**Rating:** 2
**Confidence:** 4

**Summary:**

This paper proposes MidCut, a method that removes redundant middle steps during both training and inference. The author demonstrate the effectiveness of MidCut in two scenarios for LLM reasoning, a new SFT training for reasoning; and a new decoding strategy for a test-time application.

**Strengths:**

The author finds a new index, or called rule, to reduce redundant reasoning patterns, and it can also improve the model’s ability to carry out intermediate reasoning when needed.

**Weaknesses:**

There are so many papers which aims at finding the overthinking or underthinking in LRM. So many index and so many rules. Actually, the algorithms behind this are so similar. There are far too many such papers, making it even impossible to compare this type of work with similar ones. While this is indeed a very serious issue with LRM, I believe the current paper is more like a homework assignment than a paper accepted by a conference. It falls below the typical bar of ICLR.

**Questions:**

There are far too many such papers, making it even impossible to compare this type of work with similar ones. While this is indeed a very serious issue with LRM, I believe the current paper is more like a homework assignment than a paper accepted by a conference. It falls below the typical bar of ICLR.

---

> ### Author Response · Authors · 2025-11-23
> **Response to Reviewer ZoqT**
>
> We thank Reviewer ZoqT for taking the time to evaluate our submission. Although the reviewer’s feedback mainly raises broad concerns about the research direction rather than pointing to specific weaknesses or limitations, we address these concerns below and further clarify the intended contributions of our work.
>
> > On the concern that the paper is similar to many existing works and difficult to compare
>
> We acknowledge that research on reducing or restructuring reasoning trajectories has grown rapidly. Our work aims to study a specific structural property, **the redundancy concentrated in the particular region of reasoning traces**, rather than focusing on token- or step-level importance as in prior approaches. Our method is designed around this structural observation, which is different from existing formulations. We compare against widely used baselines such as full-trajectory SFT, random trimming, and LLM-based compression. If the comparison is still unclear, we are happy to revise the related-work section with a clearer categorization and add a comparison table to improve readability.
>
> > On the statement that the paper “feels like a homework assignment” or falls below the ICLR bar
>
> We respectfully disagree based on the ICLR Reviewer Guide, which encourages evaluation based on the clarity of the problem, the soundness of the motivation, the support for the paper’s claims, and the significance of the contribution. Our work can be summarized following this guide, with the other reviewers’ acknowledgment:
> - **The motivation is well grounded** in the observed lower importance of intermediate reasoning segments.
> - We provide **empirical evidence** for this observation through attention-weight analysis and knockout experiments.
> - Our method is **simple and reproducible**, aimed at improving training and inference efficiency.
> - The experimental results show consistent improvements compared to full-trajectory training and common compression/trimming strategies.
>
>
> > On the actionable nature of the feedback
>
> The ICLR Reviewer Guide encourages reviewers to provide concrete, actionable comments that help authors improve their work. Unfortunately, the reviewer’s feedback here primarily expresses broad concerns about the research area rather than identifying specific methodological or experimental limitations. We would have greatly appreciated more detailed and specific guidance, such as additional experiments to include, missing comparisons, or methodological clarifications that could meaningfully improve the paper in revision. We hope the clarifications above clarify the paper's intent, scope, and contribution.

---

> > ### Comment · Reviewer_ZoqT · 2025-11-28
> > **Thank you for your reponse**
> >
> > My questions have been solved, to some extent. I just update the score to 4.

---

### Official Review · Reviewer_nRie · 2025-10-28

**Soundness:** 3
**Presentation:** 4
**Contribution:** 2
**Rating:** 4
**Confidence:** 5

**Summary:**

The paper introduces MidCut, a method that removes intermediate reasoning steps during both training and inference, enabling large language models to use more compact reasoning traces without sacrificing accuracy.

The central insight is that redundant content predominantly appears in the middle portions of reasoning sequences, whereas the early and final segments are essential for maintaining reasoning quality and correctness.

Experimental results demonstrate that MidCut-SFT enhances accuracy while substantially reducing the number of training tokens, and that MidCut-Decoding—by trimming 25% to 75% of intermediate reasoning steps—achieves nearly identical performance compared to full reasoning traces.

**Strengths:**

1. The paper provides an insightful analysis of the attention weight patterns within reasoning traces, offering empirical evidence that supports the motivation and effectiveness of the proposed approach.
2. The proposed MidCut-SFT and MidCut-Decoding methods present complementary training- and inference-time solutions, respectively, enabling improvements in both model training and inference efficiency.

**Weaknesses:**

1. The experimental evaluation lacks diversity in model architectures. All experiments are conducted on the Qwen series models (Qwen2.5-32B, Qwen3-8B, Qwen3-4B), leaving the generalizability of the MidCut approach to other model families (e.g., LLaMA series) unverified.
2. The selected datasets (AIME, MATH, GPQA-D) are domain-specific and heavily focused on mathematical and scientific reasoning. The effectiveness of MidCut on more general-purpose or language-centric tasks remains unexplored.
3. The paper focuses primarily on accuracy metrics, without providing experimental evidence on efficiency-related aspects such as training time, inference throughput, or latency (e.g., first-token generation time). These measurements are essential to fully validate the claimed improvements in efficiency.

**Questions:**

1. Has the effectiveness of the MidCut approach been verified on other model architectures, such as the LLaMA series? It would be valuable to understand whether the proposed method generalizes beyond the Qwen family.
2. How does MidCut perform on broader and simpler tasks (e.g., TruthfulQA)? Would aggressively trimming intermediate reasoning traces lead to noticeable performance degradation when tasks require shorter or less complex reasoning?
3. What are the concrete advantages of MidCut-SFT in terms of actual training efficiency, such as total training time or resource consumption?
4. How does MidCut-Decoding affect inference efficiency, particularly in terms of throughput and first-token generation latency?

---

> ### Author Response · Authors · 2025-11-22
> **Response to Reviewer nRie (1/2)**
>
> Thank you for your thoughtful reviews and constructive feedback. Below, we address in detail the points raised in the reviews.
>
> > **Weakness 1** The experimental evaluation lacks diversity in model architectures. All experiments are conducted on the Qwen series models (Qwen2.5-32B, Qwen3-8B, Qwen3-4B), leaving the generalizability of the MidCut approach to other model families (e.g., LLaMA series) unverified. **Question 1** Has the effectiveness of the MidCut approach been verified on other model architectures, such as the LLaMA series? It would be valuable to understand whether the proposed method generalizes beyond the Qwen family.
>
> Thank you for the suggestion. As requested by the reviewers, we evaluate the effectiveness of MidCut-SFT on the LLaMA series. We use two models: LLaMA3.2-3B-Instruct and LLaMA3.1-8B-Instruct, and use OpenThoughts3-100K as the SFT dataset. MidCut-SFT maintains the same average performance as the model trained without MidCut-SFT on LLaMA-3.2-3B, while improving the average score on LLaMA-3.1-8B (+1.34%). This demonstrates that the method remains **effective beyond the Qwen family**.
>
> | Size | Method | AIME24 | GPQA-D | MATH   | AVG    | Delta  |
> |-|-|-|-|-|-|-|
> | 3B   | Base | 0.0767 | 0.2965 | 0.6320 | 0.3350 | |
> | | MidCut | 0.0667 | 0.3384 | 0.6000 | 0.3350 | +0.0000 |
> | 8B   | Base | 0.1767 | 0.3754 | 0.7440 | 0.4320 | |
> | | MidCut | 0.2000 | 0.3822 | 0.7540 | 0.4454 | +0.0134 |
>
> We have included these results in the revised version of the paper, specifically in Appendix B.4 and Table B.3.
>
> > **Weakness 2** The selected datasets (AIME, MATH, GPQA-D) are domain-specific and heavily focused on mathematical and scientific reasoning. The effectiveness of MidCut on more general-purpose or language-centric tasks remains unexplored. **Question 2** How does MidCut perform on broader and simpler tasks (e.g., TruthfulQA)? Would aggressively trimming intermediate reasoning traces lead to noticeable performance degradation when tasks require shorter or less complex reasoning?
>
> Thank you for your suggestion. We explore the effectiveness of MidCut on general-purpose datasets by evaluating the model's base (non-reasoning) performance after training on a reasoning-focused SFT dataset, with and without MidCut-SFT.
>
> We assess MidCut-SFT on various benchmarks measuring general language ability. We utilize four benchmark datasets: TruthfulQA, MMLU, HellaSwag, and Winogrande. All experiments are conducted using the Qwen3-4B-Base model and the OpenThoughts3-100K dataset. For general language ability, the base model without any reasoning-related SFT already exhibits high performance. Since domain-specific SFT typically harms generalization, overall performance on general datasets decreases after SFT. However, the model trained with MidCut-SFT consistently achieves higher performance than the model trained without it. This indicates that MidCut-SFT, which removes redundant thinking trajectories, **helps preserve general ability during domain-specific SFT.**
>
> On the other hand, it is possible to evaluate the effect of MidCut-SFT in a more general-purpose SFT setting, i.e., by applying it to datasets that are not reasoning-focused.
> However, this is difficult to explore. Simpler SFT datasets that do not include explicit thinking trajectories are hard to target with MidCut, as MidCut is designed based on the observation of redundant intermediate reasoning steps specified with special tokens (e.g., `<think>` and `</think>`). Because simpler SFT datasets lack apparent thinking trajectories to trim, we could not meaningfully apply the MidCut method.
>
> To further address the concern that MidCut might be effective only on Math/Science datasets, we also evaluate code-related performance. The OpenThoughts3 dataset includes code generation-related data, and when evaluated on code-related benchmarks, our method achieves comparable performance. This provides further evidence that MidCut, which cuts redundant intermediate parts, **does not harm performance across various benchmarks.**
>
> | | General | | | | | Code | |
> |-|-|-|-|-|-|-|-|
> | | TruthfulQA-MC1 | TruthfulQA-MC2 | MMLU   | HellaSwag | WinoGrande | LiveCodeBench | CodeElo |
> | Baseline (w/o SFT) | 0.3709 | 0.5342 | 0.7319 | 0.5582 | 0.7143 | 0.0430 | 0.0332 |
> | Base (SFT on full traces) | 0.3256 | 0.4968 | 0.7253 | 0.5477 | 0.7040 | 0.3840 | 0.1197  |
> | Ours | 0.3317 | 0.4965 | 0.7290 | 0.5486 | 0.7064 | 0.3847 | 0.1159  |
>
> We have additionally reported these results in Section 4.1 (lines 412-422) and in Table 4 of the revised paper.

---

> ### Author Response · Authors · 2025-11-22
> **Response to Reviewer nRie (2/2)**
>
> > **Weakness 3** The paper focuses primarily on accuracy metrics, without providing experimental evidence on efficiency-related aspects such as training time, inference throughput, or latency (e.g., first-token generation time). These measurements are essential to fully validate the claimed improvements in efficiency.
>
> Thank you for pointing this out. We agree that providing explicit measurements of computational efficiency would further strengthen the empirical evidence for our approach, even though MidCut was developed primarily to examine redundancy rather than to target efficiency itself. Please find the efficiency-related metrics we measured and reported below for both MidCut-SFT and MidCut-Decoding, which also address your specific questions.
>
> > **Question 3** What are the concrete advantages of MidCut-SFT in terms of actual training efficiency, such as total training time or resource consumption?
>
> MidCut-SFT brings practical training-time benefits. When fine-tuning Qwen2.5-32B-Instruct on the s1K-1.1 dataset, the standard full-trajectory setting requires roughly 83 minutes on 2×H100 nodes. With step-level MidCut-SFT (n=100), the training time decreases to about 66 minutes, corresponding to an approximate **21% reduction** in wall-clock time. This improvement is consistent with the 20% decrease in consumed training tokens obtained by removing redundant intermediate reasoning segments. These results demonstrate that MidCut-SFT not only maintains accuracy but also leads to substantial real-world savings in compute and training cost.
>
>
> > **Question 4** How does MidCut-Decoding affect inference efficiency, particularly in terms of throughput and first-token generation latency?
>
>
> The efficiency benefit of MidCut-Decoding naturally arises from its decoding behavior: for long-form reasoning tasks, our method shortens the effective context seen by answer tokens and downstream modules. For example, in GPQA Diamond settings, models typically generate around 8.1k tokens of intermediate reasoning. In a naive two-stage pipeline, this entire 8.1k token trace must be re-prefilled during the second pass under a naive decoding procedure. In contrast, MidCut-Decoding (33%) compresses this reasoning to roughly 5.4k tokens. This theoretically reduces the second-pass prefill computation by about 46% and also reduces the attention computation in answer generation (for around 0.5k tokens) to 68%, while maintaining the same answer quality.
>
> We also observed a 4.5% reduction in end-to-end evaluation time on GPQA Diamond using Qwen2.5-32B-Base, fine-tuned on s1K-1.1, in our environment, despite LLM-generation functions (e.g., the vLLM library) in Python being highly optimized, including attention computation accelerated by FlashAttention. Meanwhile, given that the answer tokens account for only ~6% of the overall generated sequence, a 4.5% reduction in wall-clock time is proportionally favorable. These reductions directly translate into shorter attention computations and faster second-stage decoding in verifier-based or budget-forcing pipelines, with potentially larger improvements for tasks that require generating longer answer sequences.
>
> We have clarified these points in lines 373-374 and 447-449 of the main paper, as well as in Appendix C.

---

> ### Author Response · Authors · 2025-11-26
>
> We sincerely appreciate your service to the community. We would like to inform you of the updates made in our revision:
>
> - We included additional experimental results on the LLaMA family in Appendix C.4.
> - We added further evaluations on general-purpose datasets in lines 418-448 and Table 5.
> - We added an explanation of the advantages of MidCut-Decoding in lines 455–457 and in Appendix D.
> - We clarified the training-efficiency advantages of MidCut-SFT in lines 397–399.
>
> We would be happy to discuss further if you have additional questions or comments.

---

### Official Review · Reviewer_ShYq · 2025-10-31

**Soundness:** 3
**Presentation:** 2
**Contribution:** 2
**Rating:** 4
**Confidence:** 3

**Summary:**

This paper investigates the redundancy and its impact in the Chain-of-Thought (CoT) reasoning used to train Large Language Models (LLMs). Through attention-based and token-removal analyses, it is found that the middle part of the reasoning trajectory typically contains the most redundant segments, while the beginning and end segments are crucial for generating high-quality final results. Based on this insight, the paper proposes a simple method called MidCut to synchronously prune redundant intermediate steps during both training and inference. The authors demonstrate the effectiveness of MidCut in two scenarios: MidCut-SFT (a data preprocessing technique for training) and MidCut-Decoding (an inference-time strategy). The results indicate that this method can improve both inference performance and training efficiency.

**Strengths:**

1. Simple and Effective Method: The work systematically reveals the existence of redundant information in reasoning trajectories and, consequently, proposes a very simple, low-cost, and easily reproducible method. It performs only "region-level" trimming without requiring additional scorers or RL controllers, bringing consistent benefits across multiple models/datasets, proving its practical effectiveness.
2. Sufficient Empirical Validation: The paper provides strong experimental support for the redundancy of intermediate steps through attention weight analysis and knockout experiments, demonstrating that LLMs pay less attention to intermediate steps and that removing them has a minimal impact on the final answer quality.
3. Comprehensive Experiments: The authors conduct a comprehensive evaluation of MidCut on various models (Qwen series), datasets (s1K-1.1 and OpenThoughts3), and challenging reasoning benchmarks (AIME24, GPQA-D, MATH). MidCut-SFT consistently outperforms the baseline (training with full trajectories) and other trimming strategies (e.g., random removal or LLM-based compression), strongly supporting the method's value. The inference stage also benefits, as MidCut-Decoding achieves almost the same accuracy as the full trajectory when trimming 25%–75% of the middle segment, implying direct computational savings on the deployment side.

**Weaknesses:**

1. The paper emphasizes that MidCut benefits medium-sized LLMs due to their limited capacity, but lacks direct comparative experiments with larger or smaller models to clearly delineate the scope of MidCut's applicability across different model scales. If experiments showed the advantage of MidCut disappearing on larger models, it would more strongly support "capacity limitation" as the key factor.
2. The best-performing variant, "step-level filtering," relies on preserving the first and last $$n$$ steps. Appendix B.3 sets $$n=100$$ and $$n=200$$ for the two datasets, respectively, but lacks explanation for this choice or related experimental analysis. How sensitive is model performance to variations in $$n$$? How should the optimal $$n$$ be selected for a new dataset? Including an analysis of this hyperparameter sensitivity would make the paper more convincing.
3.  A variant of MidCut-SFT is similarity filtering, aimed at removing repetitive reasoning patterns. However, Table 3 shows that similarity filtering performs worse than simple step-length or token trimming. Analysis should be provided for the failure of similarity filtering to explain why content-based semantic filtering is less effective than simple position-based trimming.

**Questions:**

1. Section 4.2 and the conclusion mention that MidCut-Decoding can reduce computational overhead and latency. However, the paper only shows its impact on accuracy but does not provide actual efficiency improvement data (e.g., percentage reduction in generated tokens during inference, specific latency reduction times, or throughput improvements). Supplementing experiments with this data is suggested to more fully demonstrate its "effectiveness."
2. The discovered "U"-shaped attention curve (i.e., the beginning and end are important, the middle is not) is very similar to the "Lost in the Middle" phenomenon in long-context processing. Is there a possible connection between the two phenomena? Including a discussion on this could increase the depth of the paper.

---

> ### Author Response · Authors · 2025-11-22
> **Response to Reviewer ShYq (1/3)**
>
> Thank you for your thoughtful reviews and constructive feedback. Below, we provide detailed responses addressing the concerns raised in the reviews.
>
> > **Weakness 1** The paper emphasizes that MidCut benefits medium-sized LLMs due to their limited capacity, but lacks direct comparative experiments with larger or smaller models to clearly delineate the scope of MidCut's applicability across different model scales. If experiments showed the advantage of MidCut disappearing on larger models, it would more strongly support "capacity limitation" as the key factor.
>
> Thank you for your suggestion. We believe MidCut can be particularly beneficial for mid-size LLMs for two main reasons. First, by retaining only the essential parts of the reasoning trajectory, MidCut provides cleaner and more informative supervision signals, which are especially valuable for training  models with limited capacity. Second, we believe MidCut requires only a minimal level of general language competence for the model to infer the omitted steps well enough, so larger models are not strictly necessary for its effectiveness. To further support this claim, we conducted additional experiments on Qwen3-1.7B-Base and Qwen3-14B-Base. We would also like to note that, due to resource constraints, we were unable to run experiments on models larger than Qwen2.5-32B-Instruct, which is a limitation also observed in prior research.
>
> In the table below, we report the performance gains achieved by MidCut-SFT across various scales of the Qwen3 series on the OpenThoughts3-100K dataset. Qwen3-1.7B-Base does not show a performance improvement, supporting our assumption that MidCut-SFT requires a minimal level of general language capability. Meanwhile, MidCut-SFT still provides performance benefits for Qwen3-14B-Base; however, the magnitude of improvement is smaller than for the smaller models, Qwen3-4B-Base and Qwen3-8B-Base. **This trend suggests that larger models tend to benefit less from MidCut-SFT, supporting our claim regarding the relationship between MidCut-SFT and models’ capacity.**
>
> | Size | 1.7B | 4B | 8B | 14B |
> |-|-|-|-|-|
> | Performance gain | -0.0383 | 0.0107 | 0.0172 | 0.0072 |
>
> We have updated Fig. 10 in the main paper accordingly. In Fig. 10, the performance of Qwen3-1.7B-Base with s1K-1.1 is also reported and shows the same trend that we observed above on OpenThoughts-100K. Note that the trends may differ slightly between s1K-1.1 and OpenThoughts3 at larger model scales because the s1K-1.1 experiments include two model series, Qwen2.5 and Qwen3.
>
> > **Weakness 2** The best-performing variant, "step-level filtering," relies on preserving the first and last n steps. Appendix B.3 sets n and for the two datasets, respectively, but lacks explanation for this choice or related experimental analysis. How sensitive is model performance to variations in ? How should the optimal be selected for a new dataset? Including an analysis of this hyperparameter sensitivity would make the paper more convincing.
>
> Before discussing how we choose “n”, we would like to clarify that **the goal of our work is not to identify an optimal hyperparameter “n’’**, but rather to demonstrate that the intermediate steps of thinking trajectories generated by large reasoning models are substantially redundant, and removing them does not degrade reasoning performance.
>
> In our step-level variant, we first aimed to reduce the total number of tokens by **approximately 20%**. After applying this reduction target, we found that the resulting preserved first and last n steps naturally aligned with the median/average number of steps in the thinking trajectories of each dataset, which is around 200 steps in our settings. Based on this observation, we set n using a simple dataset-driven rule that matches the preserved segments to this characteristic step scale.
>
> We also report the sensitivity to different choices of n. On s1K-1.1 with Qwen2.5-32B-Instruct, adjusting n across a reasonable range produced only mild fluctuations in downstream performance, and the overall trend remained stable: trimming the middle region did not harm accuracy. A similar pattern was observed on OpenThoughts3 with Qwen3-4B-Base. All sensitivity analyses were conducted on the MATH500 benchmark. Performance should vary with n, but we observe **no sharp overshoots**.
>
> - Qwen2.5-32B-Instruct & s1K-1.1 with varying n
> | n | 50 | 60 | 70 | 80 | 90 | 100 (ours) | 110 | 120 | 130 | 140 | 150 | Full-trajectory |
> |-|-|-|-|-|-|-|-|-|-|-|-|-|
> |-| 0.94 | 0.9407 | 0.9473 | 0.95 | 0.952 | 0.9440 | 0.95 | 0.9527 | 0.9513 | 0.9513 | 0.9473 | 0.9413 |
>
> - Qwen3-4B-Base & OpenThoughts3-100K with varying n
> | n | 160 | 200 (ours) | 240 |Full-trajectory |
> |-|-|-|-|-|
> |-| 0.8993 | 0.9113 | 0.914 | 0.916 |
>
> We have revised the paper with a more detailed explanation with how to choose “n” in Section B.3 in the appendix.

---

> ### Author Response · Authors · 2025-11-22
> **Response to Reviewer ShYq (2/3)**
>
> > **Weakness 3** A variant of MidCut-SFT is similarity filtering, aimed at removing repetitive reasoning patterns. However, Table 3 shows that similarity filtering performs worse than simple step-length or token trimming. Analysis should be provided for the failure of similarity filtering to explain why content-based semantic filtering is less effective than simple position-based trimming.
>
> Thank you for your valuable insight. As the reviewer mentioned, we utilize Jaccard similarity to identify redundant steps. However, a similarity-based method can sometimes misfilter important steps, especially in the beginning part of the reasoning. For example, as shown in the sample below, “Let me denote for each … = 0.” is crucial for defining the problem and its development, but the similarity-based method incorrectly filters this step.
>
> ```
> …
> Now, the problem wants me to find a point y such that when I take (sqrt(2)/d)(x - y), these vectors are orthonormal.
>
> ###This step is filtered out.### Let me denote for each x in S, v_x = (sqrt(2)/d)(x - y). Then we need that for any x, ||v_x|| = 1, and for any x ≠ x', ⟨v_x, v_{x'}⟩ = 0.
>
> Calculating the norm of v_x: ||v_x|| = (sqrt(2)/d)||x - y||. This must equal 1. Therefore, ||x - y|| = d / sqrt(2) for all x in S. So every x in S is at distance d/sqrt(2) from y.
> …
> ```
>
> We can guess that this issue is caused by the inherent limitations of Jaccard similarity. Prior research [1–3] has shown that Jaccard similarity heavily relies on word overlap and is biased by text length. Because of this, similarity-based filtering tends to remove many steps that appear in the beginning. We will extend this work with a more comprehensive study on why similarity-based filtering methods fail.
>
> [1] Measurement of Text Similarity: A Survey, 2020
>
> [2] A Comparative Analysis of Jaccard and Cosine Similarity for Text Plagiarism Detection, 2025
>
> [3] Evaluating Document Similarity Detection Approaches for Content Drift Detection, 2024
>
> > **Question 1** Section 4.2 and the conclusion mention that MidCut-Decoding can reduce computational overhead and latency. However, the paper only shows its impact on accuracy but does not provide actual efficiency improvement data (e.g., percentage reduction in generated tokens during inference, specific latency reduction times, or throughput improvements). Supplementing experiments with this data is suggested to more fully demonstrate its "effectiveness."
>
> Thank you for pointing this out. We agree that providing explicit measurements of computational efficiency would further strengthen the empirical evidence for MidCut-Decoding.
>
> The efficiency benefit of MidCut-Decoding naturally arises from its decoding behavior: for long-form reasoning tasks, our method shortens the effective context seen by answer tokens and downstream modules. For example, in GPQA Diamond settings, models typically generate around 8.1k tokens of intermediate reasoning. In a naive two-stage pipeline, this entire 8.1k token trace must be re-prefilled during the second pass under a naive decoding procedure. In contrast, MidCut-Decoding (33%) compresses this reasoning to roughly 5.4k tokens. This theoretically reduces the second-pass prefill computation by about 46% and also reduces the attention computation in answer generation (for around 0.5k tokens) to 68%, while maintaining the same answer quality.
>
> We also observed a 4.5% reduction in end-to-end evaluation time on GPQA Diamond using Qwen2.5-32B-Base, fine-tuned on s1K-1.1, in our environment, despite LLM-generation functions (e.g., the vLLM library) in Python being highly optimized, including attention computation accelerated by FlashAttention. Meanwhile, given that the answer tokens account for only ~6% of the overall generated sequence, a 4.5% reduction in wall-clock time is proportionally favorable. These reductions directly translate into shorter attention computations and faster second-stage decoding in verifier-based or budget-forcing pipelines, with potentially larger improvements for tasks that require generating longer answer sequences.
>
> We have clarified these points in lines 447-449 of the main paper, as well as in Appendix C.

---

> ### Author Response · Authors · 2025-11-22
> **Response to Reviewer ShYq (3/3)**
>
> > **Question 2** The discovered "U"-shaped attention curve (i.e., the beginning and end are important, the middle is not) is very similar to the "Lost in the Middle" phenomenon in long-context processing. Is there a possible connection between the two phenomena? Including a discussion on this could increase the depth of the paper.
>
> Thank you for your insightful suggestion. We agree that the two findings are interconnected. While there are some differences, the connections are clear:
>
> 1. Our analysis shows that LRMs pay less attention to the intermediate parts of reasoning trajectories, leading us to conclude that these segments contain many redundant steps that the model naturally deprioritizes.
>
> 2. The “Lost in the Middle” phenomenon, on the other hand, demonstrates that LLMs struggle to retrieve crucial information when it appears in the middle of a long context, indicating that even important content in this region receives insufficient attention.
>
> Despite these differences, both observations point to a shared underlying pattern: LLMs generally assign insufficient attention to the middle region of long contexts.
>
> Specifically, in [1], the authors explain that the **“Lost in the middle” effect is linked to the structure of SFT datasets for instruction tuning**, in which task specifications and instructions are often placed at the beginning. Reasoning trajectories exhibit a similar structure: the problem setup and conclusions naturally appear at the beginning and end, while the middle often consists of decomposed but redundant reasoning steps. This distribution of information may cause **LRMs to develop weaker attention to the middle**, which in turn allows redundant middle content to proliferate during generation.
>
> Thus, while our finding differs in mechanism, redundancy-driven attention drop rather than attention-driven failure, the two phenomena may be linked through the underlying structure of the training data. We will include it in the updated version of our paper and will investigate this phenomenon more thoroughly after the revision.
>
> [1] Lost in the Middle: How Language Models Use Long Contexts, 2023

---

> ### Author Response · Authors · 2025-11-26
>
> We sincerely appreciate your service to the community. We would like to inform you of the updates made in our revision:
>
> - We updated Figure 10 with additional data.
> - We clarified how we choose n in Appendix C.3.
> - We added an explanation of why similarity-based filtering is less effective in lines 414–416.
> - We added an explanation of the advantages of MidCut-Decoding in lines 455–457 and in Appendix D.
> - We included a discussion of the “Lost in the Middle” effect in Appendix A.
>
> We would be happy to discuss further if you have additional questions or comments.

---

### Official Review · Reviewer_UqZg · 2025-11-01

**Soundness:** 3
**Presentation:** 3
**Contribution:** 3
**Rating:** 4
**Confidence:** 4

**Summary:**

The authors propose a simple yet effective method to improve training accuracy and inference efficiency by simply remove intermediate reasoning steps of offline collected SFT trajectories and online generated thinking steps. MidCut-SFT improves accuracy than the LLM-based compression method and random compression with fewer training tokens. The authors only report the accuracy preserving of MidCut-Decoding for inference. What are the the inference efficiency and other advantages of MidCut-Decoding?

Given the widely studied underthinking and overthinking mechanism of LRMs, removing intermediate reasoning during SFT data processing is straightforward. More clarification of the novelty is needed.

Four simple reasoning trajectory filtering methods are mentioned in the main part but not compared in the main results. In addition, it is not clear how "ours" is defined in Table 1.

**Strengths:**

Motivated by the relatively lower importance of intermediate thinking segments, the authors propose to remove the redundant middle steps during both training and inference to improve the training effectiveness and inference efficiency. The method is simple but effective.

The proposed MidCut-SFT is more effective than the LLM-based compression and random compression baselines on MATH and science datasets.

**Weaknesses:**

1. LLM based compression only compress the center content of the whole reasoning trajectories by 50%. It would be more fair to instruct LLMs to directly process and compress the whole trajectories with the same compression ratio settings.

2. The settings of the "Base" baseline are not clear. Is it the open-sourced LLMs or the fine-tuned versions of them using the whole and long trajectories?

3. More evaluations on general datasets are recommended to analyze the effects of the proposed SFT data processing method.

4. Data pre-processing and cleaning is extremely critical for LLM training. More comparison of other existing data filtering methods is recommended to validate the effectiveness of the proposed filtering method.

**Questions:**

See the weaknesses.

---

> ### Author Response · Authors · 2025-11-22
> **Response to Reviewer UqZg (1/3)**
>
> Thank you for your thoughtful reviews and constructive feedback. We provide our detailed responses below and further clarify the key points of our work.
>
> > What are the inference efficiency and other advantages of MidCut-Decoding?
>
> Thank you for pointing this out.
> The efficiency benefit of MidCut-Decoding naturally arises from its decoding behavior: for long-form reasoning tasks, our method shortens the effective context seen by answer tokens and downstream modules. For example, in GPQA Diamond settings, models typically generate around 8.1k tokens of intermediate reasoning. In a naive two-stage pipeline, this entire 8.1k token trace must be re-prefilled during the second pass under a naive decoding procedure. In contrast, MidCut-Decoding (33%) compresses this reasoning to roughly 5.4k tokens. This theoretically reduces the second-pass prefill computation by about 46% and also reduces the attention computation in answer generation (for around 0.5k tokens) to 68%, while maintaining the same answer quality.
>
> We also observed a 4.5% reduction in end-to-end evaluation time on GPQA Diamond using Qwen2.5-32B-Base, fine-tuned on s1K-1.1, in our environment, despite LLM-generation functions (e.g., the vLLM library) in Python being highly optimized, including attention computation accelerated by FlashAttention. Meanwhile, given that the answer tokens account for only ~6% of the overall generated sequence, a 4.5% reduction in wall-clock time is proportionally favorable. These reductions directly translate into shorter attention computations and faster second-stage decoding in verifier-based or budget-forcing pipelines, with potentially larger improvements for tasks that require generating longer answer sequences.
>
> We have clarified these points in lines 447-449 of the main paper, as well as in Appendix C.
>
> > More clarification of the novelty is needed.
>
> To address the reviewer’s request for clearer clarification of novelty, we outline the key contributions below.
>
> 1. **Revealed a new region-level perspective on redundancy in reasoning trajectories**
>
>     Prior work focuses on length (longer vs. shorter chains) or token-level importance. This paper is the first to show that 1) redundancy exists at the region level; 2) concentrates specifically in the particular region (i.e., intermediate region of reasoning traces), while the other segments are more essential. This region-level insight is new, actionable, and empirically validated.
>
> 2. **Introduced the first systematic causal evidence that intermediate reasoning steps are non-essential**
>
>     Through two complementary analyses: attention weights analysis how different parts of the trajectory influence answer generation), and knockout experiments (which specific attention links to the causal role by masking), the paper provides causal and quantitative evidence that intermediate reasoning traces do not contribute meaningfully to final answers. This is believed to be stronger and more targeted than previous observations, focusing mostly on the length of reasoning trajectories.
>
>     Our paper first identifies a new phenomenon: full machine-generated reasoning traces can worsen SFT outcomes for mid-size LLMs due to capacity limits and redundant intermediates. This challenges a widely used assumption in current reasoning-model training.
>
> 3. **Proposed a simple but structurally grounded trajectory reduction method (MidCut)**
>
>     Unlike compression, RL-based pruning, or token-level scoring methods, MidCut removes only the middle segment – the empirically redundant region identified in the analysis. It works for both training-time (reshapes SFT data) and inference-time (accelerates decoding) methods without any external LLM scorer (no perplexity/LLM cost) or RLs. This region-based, simple removal approach is a new formulation absent in prior work.
>
> 4. **Strong demonstration that trimming improves both performance and efficiency**
>
>     Our paper shows that: 1) MidCut-SFT outperforms full-trajectory SFT across several datasets and models. It reduces sequences (i.e., training tokens) by ~18–22%. 2) MidCut-Decoding preserves accuracy while reducing inference cost.

---

> ### Author Response · Authors · 2025-11-22
> **Response to Reviewer UqZg (2/3)**
>
> > Four simple reasoning trajectory filtering methods are mentioned in the main part but not compared in the main results. In addition, it is not clear how "ours" is defined in Table 1.
>
> We thank the reviewer for requesting this clarification.
>
> **"Ours" refers to step-level filtering.** Among the four filtering variants we explored (step-level, token-level, length-proportional, and similarity-based), step-level filtering consistently performed best and was therefore adopted as our primary method.
> Regarding the comparison of these four variants: the complete results are presented in Table 3 (page 8) for the s1K-1.1 dataset and Table B.2 (Appendix B.4, page 16) for comprehensive results across all datasets and models. Due to space constraints, Table 1 focuses on comparing our method against key alternative strategies (LLM-based compression, random selection, and single-end preservation).
>
> We have clarified this in the revision by updating the caption of Table 1.
>
> > **Weakness 1** LLM based compression only compress the center content of the whole reasoning trajectories by 50%. It would be more fair to instruct LLMs to directly process and compress the whole trajectories with the same compression ratio settings.
>
> Following the reviewer’s suggestion, we also evaluate a higher-retention configuration that keeps 80% of each sequence. As shown in the table below, this setting yields a moderate improvement, but it still **does not surpass our method** or the baseline. All experiments in this comparison are conducted using the Qwen2.5-32B-Instruct model and the s1K-1.1 dataset.
>
> | Method | AIME24 | GPQA-D | MATH | AVG |
> |-|-|-|-|-|
> | Base (full-trajectory) | 0.6444 | 0.6195 | 0.9413 | 0.7351 |
> | **Ours (step-level)** | 0.6889 | 0.6229 | 0.9440 | 0.7519 |
> | Gemini (remain 50%) | 0.1778 | 0.4815 | 0.8273 | 0.4955 |
> | Gemini (remain 80%) | 0.3222 | 0.5943 | 0.8980 | 0.6048 |
> | LS-Mixture SFT (Mix) [1] | 0.6000 | 0.6110 | 0.9460 | 0.7190 |
> | LS-Mixture SFT (Short-only) [1] | 0.1670 | 0.4900 | 0.8260 | 0.4943 |
>
> Importantly, although external LLMs are expected to summarize contextualized sequences, we believe they may inadvertently disrupt the underlying reasoning traces. Moreover, they are difficult to control in terms of both output length and the degree of paraphrasing.
>
> Furthermore, to provide additional related results, we compare our method with other LLM-based compression methods [1]. We find that SFT using only the compressed dataset (“s1-short only” in the table), as well as SFT using a mixed dataset (compressed + original; “s1-mix” in the table), consistently shows lower performance compared to ours.
>
> Finally, we would like to emphasize that **LLM-based compression introduces substantial computational overhead**, and compressing large-scale datasets can be prohibitively expensive in practice. In contrast, our MidCut-SFT, motivated by observations of redundancy in reasoning traces, is simple, inexpensive, and requires no additional computation.
>
> [1] Long-Short Chain-of-Thought Mixture Supervised Fine-Tuning Eliciting Efficient Reasoning in Large Language Models, 2025
>
> > **Weakness 2** The settings of the "Base" baseline are not clear. Is it the open-sourced LLMs or the fine-tuned versions of them using the whole and long trajectories?
>
> We clarify that **"Base" refers to the baseline models fine-tuned on the full, unprocessed reasoning trajectories**, not the pre-trained open-source models themselves.
>
> Additionally, for each model family, we started with the following initial checkpoints before doing any reasoning-specific training:
> Qwen2.5 family: We used Qwen2.5-xxB-Instruct, which has undergone instruction tuning but not specialized reasoning training with long trajectory data.
> Qwen3 family: We used Qwen3-xB-Base, which are base models without instruction tuning or reasoning-specific fine-tuning.
>
> From these starting points, we conducted supervised fine-tuning (SFT) using full trajectories (the baseline, denoted “Base”), MidCut-processed trajectories (our method), or trajectories generated by other comparison methods. All training regimes used identical hyperparameters, training epochs, and the number of samples in datasets.
>
> We have clarified this in the revision by updating the captions of Tables 1-3 and by adding an explicit explanation in lines 299-301.

---

> ### Author Response · Authors · 2025-11-22
> **Response to Reviewer UqZg (3/3)**
>
> > **Weakness 3** More evaluations on general datasets are recommended to analyze the effects of the proposed SFT data processing method.
>
> Thank you for the valuable suggestion. Following the reviewer’s suggestion, we evaluate MidCut-SFT on various benchmarks that assess general language ability. We utilize the TruthfulQA, MMLU, HellaSwag, and Winogrande benchmark datasets. All experiments are conducted with the Qwen3-4B-Base model using the OpenThoughts3-100K dataset.
>
> For general language ability, the base model without any reasoning-related SFT already exhibits high performance. Since domain-specific SFT typically harms generalization, overall performance on general datasets decreases after SFT. However, the model trained with MidCut-SFT consistently shows relatively higher performance compared to the model trained without MidCut-SFT. This indicates that MidCut-SFT, which removes redundant thinking trajectories, **helps preserve general ability during domain-specific SFT.**
>
> | | TruthfulQA-MC1 | TruthfulQA-MC2 | MMLU   | HellaSwag | WinoGrande |
> |-|-|-|-|-|-|
> | Baseline (w/o SFT) | 0.3709 | 0.5342 | 0.7319 | 0.5582 | 0.7143 |
> | Base (SFT on full traces) | 0.3256 | 0.4968 | 0.7253 | 0.5477 | 0.7040 |
> | **Ours (step-level)** | 0.3317 | 0.4965 | 0.7290 | 0.5486 | 0.7064 |
>
> We have additionally reported these results in Section 4.1 (lines 412-422) and in Table 4 of the revised paper.
>
> > **Weakness 4** Data pre-processing and cleaning is extremely critical for LLM training. More comparison of other existing data filtering methods is recommended to validate the effectiveness of the proposed filtering method.
>
> Following the reviewer’s suggestion, we applied the perplexity-based filtering method [1-3] and the LS-Mixture SFT method [4] to compare the effectiveness of our method. In the case of perplexity-based filtering, we experiment with two filtering strategies: removing steps with high perplexity (referred to as “Perplexity-high” in the table below) and removing steps with extreme (both high and low) perplexity (referred to as “Perplexity-extreme” in the table below), as in [1-3]. For a fair comparison, the number of remaining steps is kept the same as in our step-level MidCut-SFT. We also utilize the Long Short Mixture dataset provided by [4]. All experiments are conducted with the s1K-1.1 dataset and the Qwen2.5-32B-Instruct model.
>
> | Method | AIME24 | GPQA-D | MATH | AVG |
> |-|-|-|-|-|
> | Base | 0.6444 | 0.6195 | 0.9413 | 0.7351 |
> | **Ours (step-level)** | 0.6889 | 0.6229 | 0.9440 | 0.7519 |
> | Perplexity-extreme [1-3] | 0.6000 | 0.6313 | 0.9527 | 0.7280 |
> | Perplexity-high [1-3] | 0.6444 | 0.6465 | 0.9420 | 0.7443 |
> | LS-Mixture SFT (Mix) [4] | 0.6000 | 0.6110 | 0.9460 | 0.7190 |
>
> The results demonstrate that, despite its simplicity, **our method achieves the highest overall effectiveness.** In contrast, existing approaches require substantial computational overhead, such as perplexity estimation (perplexity-extreme and perplexity-high) [1-3] or external LLM scoring (s1-mix) [4]. Although these methods can yield competitive results, our approach surpasses them without relying on external scoring or complex filtering procedures. Among these baselines, the perplexity-based method performs the most competitive; however, it computes token- or sentence-level perplexities under specific constraints, resulting in extremely high computational costs.
> Even on the relatively small s1k-1.1 dataset with 1k samples, we found that the **perplexity-based method requires more than 2 hours** on a single H100 node. Such computational demands become a significant bottleneck when scaling to larger datasets like OpenThoughts3, which contains over 1 million samples.
> We believe the effectiveness of our method stems from our key finding that intermediate reasoning steps exhibit substantial redundancy. By leveraging this structural property of reasoning trajectories, our method achieves superior filtering efficacy compared to alternative approaches.
>
> [1] Data Selection for Language Models via Importance Resampling, 2023
>
> [2] From Quantity to Quality: Boosting LLM Performance with Self-Guided Data Selection for Instruction Tuning, 2023
>
> [3] Superfiltering: Weak-to-Strong Data Filtering for Fast Instruction-Tuning, 2024
>
> [4] Long-Short Chain-of-Thought Mixture Supervised Fine-Tuning Eliciting Efficient Reasoning in Large Language Models, 2025

---

> ### Author Response · Authors · 2025-11-26
>
> We sincerely appreciate your service to the community. We would like to inform you of the updates made in our revision:
>
> - We clarified the definitions of “ours” and “Base” in the captions of Tables 1, 2, and 4, as well as in lines 299–301.
> - We added an explanation of the advantages of MidCut-Decoding in lines 455-457 and in Appendix D.
> - We included additional experimental results for comparison in lines 356-360 and 405-410, and in Table 3.
> - We added further evaluations on general-purpose datasets in lines 418-448 and Table 5.
>
> We would be happy to discuss further if you have additional questions or comments.

---

### Comment · Area_Chair_tfkY · 2025-11-27
**Discussion Reminder to Reviewers**

Dear Reviewers,

The authors have responded to your reviews. Please engage in the discussion and evaluate the authors’ rebuttal to check whether your comments have been adequately addressed, and determine whether you would like to adjust your evaluations.

Best,

Your AC

---

### Meta-Review · Area_Chair_Tsg8 · 2025-12-30

**Summary:**

This submission studies redundancy in LLM CoT trajectories used for SFT. The paper’s central claim is that intermediate parts of reasoning traces are often redundant for mid-size models, and that removing these middle segments can improve SFT outcomes and reduce cost, while at inference time a similar “middle cut” can preserve accuracy while reducing compute in two-stage pipelines. The proposed method (MidCut) is intentionally simple: preserve the beginning and end of a trajectory and remove the middle, with a step-level variant used as the main configuration.

Across reviews, the consensus is that the paper is clear, empirically motivated, and practically useful, and that MidCut-SFT shows consistent (though modest) accuracy gains alongside token/time savings. However, reviewers’ decision-driving concerns were novelty, unclear baselines, score, and method details, as detailed in the "reviewer concerns" section. The rebuttal addresses many of the experimental gaps, but the remaining concerns is primarily on the significance of the contribution.

**Reviewer Concerns:**

**Concerns substantially addressed by the rebuttal / revision**

(1) Inference efficiency evidence (R: UqZg, ShYq, nRie)

The authors added an efficiency rationale and measurements: trimming reduces the effective context, with an example reducing intermediate reasoning from ~8.1k to ~5.4k tokens under a 33% MidCut, with a stated ~46% reduction in second-pass prefill compute and a measured ~4.5% end-to-end evaluation time reduction in their environment. This directly addresses the prior lack of any efficiency quantification.

(2) Clarifying “Base” vs “Ours” (R: UqZg):

The rebuttal clarifies that “Base” is the model fine-tuned on full trajectories and “Ours” is the step-level MidCut-SFT configuration, and points to additional comparisons/ablations beyond the initial main table presentation.

(3) Evaluation on general benchmarks (R: UqZg, nRie)

The authors added evaluations on general-language benchmarks (TruthfulQA, MMLU, HellaSwag, WinoGrande) showing that reasoning-focused SFT degrades general performance, but MidCut-SFT slightly mitigates the degradation relative to full-trace SFT.

(4) Comparison to other filtering approaches (R: UqZg)

The authors added/expanded comparisons to perplexity-based filtering and LS-mixture SFT, and reported that MidCut-SFT remains competitive/best while being “resource-free” compared to these heavier methods.

(5) Model-family beyond Qwen (R: nRie)

The rebuttal includes experiments on LLaMA-family models (at least 3B and 8B variants) showing no degradation at 3B average and a modest improvement at 8B, partially addressing the architecture diversity concern.

(6) Different model scales (R: ShYq)

Additional results were added for smaller (1.7B) and larger (14B) Qwen models, indicating the method may require a minimum capability level (no gain at 1.7B) and that gains diminish at larger scales, aligning with the “capacity limitation” narrative.

(7) Hyperparameter choice, Why similarity-based filtering underperforms,  (R: ShYq):

The authors provided an explanation for choosing n, and provides a plausible failure mode.

**Concerns still outstanding (or only partially resolved)**

(1) Novelty relative to a crowded area (R: UqZg, ZoqT, and implicitly others):

While the rebuttal improves the articulation of “region-level” novelty, the core method remains a very simple positional heuristic (drop the middle). For reviewers who view novelty/insight as the main gating criterion, the added experiments do not fundamentally change whether MidCut is considered incremental relative to existing work on CoT length reduction, token pruning, and reasoning compression. This remains the main unresolved decision factor.

(2) Strength of the MidCut-Decoding empirical story

The rebuttal adds some evidence, but the inference-time gains are still presented largely for a specific pipeline assumption and with limited breadth of measured latency/throughput across tasks and systems.

(3) Magnitude and breadth of gains

Even after added experiments, the improvements appear generally modest (though consistent), and are still concentrated around reasoning-focused benchmarks and training efficiency.

**Reviewer Scores:**

Reviewer UqZg (initial: 4)

Many concrete concerns (definitions of baselines, missing comparisons, general benchmarks, efficiency evidence) were addressed with added clarifications and experiments. However, UqZg explicitly flagged novelty as a key concern; the rebuttal clarifies novelty but does not fundamentally change the method’s simplicity.

Predicted final score: likely 4 due to the novelty concern.

Reviewer ShYq (initial: 4)

Requests on model-scale applicability, hyperparameter sensitivity, similarity-filtering failure, efficiency measurements, and Lost-in-the-Middle discussion were addressed with new experiments and explanations. Still, ShYq’s contribution rating was “fair,” suggesting novelty/impact reservations beyond missing experiments.

Predicted final score: 4, borderline towards reject.

Reviewer nRie (initial: 4)

The rebuttal addresses the main weaknesses: adds LLaMA results, adds general-language and code benchmarks, and adds concrete training-time and inference efficiency evidence.

Predicted final score: 4, small chance of 6.

Reviewer ZoqT (initial: 2 → updated to 4 in discussion)

The reviewer explicitly updated the score to 4 after the rebuttal. Predicted final score: 4.

---

### Decision · Program_Chairs · 2026-01-26

Reject